# Metagenomic insights into the metabolism of microbial communities that mediate iron and methane cycling in Lake Kinneret iron-rich methanic sediments

Michal Elul[1], Maxim Rubin-Blum[2], Zeev Ronen[3], Itay Bar-Or[4], Werner Eckert[5], Orit Sivan[1]

5  [1] Department of Earth and Environmental Science, Ben-Gurion University of the Negev, Beer Sheva, Israel

[2] Israel Limnology and Oceanography Research, Haifa, Israel

[3] Department of Environmental Hydrology and Microbiology, The Zuckerberg Institute for Water Research, Ben-Gurion University of the Negev, Sede-Boqer, Israel

[4] Central Virology Laboratory, Ministry of Health, Sheba Medical Center, Ramat Gan, Israel

10  [5] Israel Oceanographic & Limnological Research, The Yigal Allon Kinneret Limnological Laboratory, Migdal, Israel

*Correspondence to* Michal Elul (elul@post.bgu.ac.il)

**Abstract.** Complex microbial communities facilitate iron and methane transformations in anoxic methanic sediments of freshwater lakes, such as Lake Kinneret (The Sea of Galilee, Israel). The phylogenetic and functional diversity of these consortia are not fully understood, and it is not clear which lineages perform iron reduction and anaerobic oxidation of methane (AOM). Here, we investigated microbial communities from both natural Lake Kinneret iron-rich methanic sediments (>20 cm depth) and iron-amended slurry incubations from this zone using metagenomics, focusing on functions associated with iron reduction and methane cycling. Analyses of the phylogenetic and functional diversity indicate that consortia of archaea (mainly Bathyarchaeia, Methanomicrobia, Thermoplasmata, and Thermococci) and bacteria (mainly Chloroflexi (Chloroflexota), Nitrospirae (Nitrospirota) and Proteobacteria) perform key metabolic reactions such as amino acid uptake and dissimilation, organic matter fermentation, and methanogenesis. The Deltaproteobacteria, especially Desulfuromondales (Desulfuromonadota), have the potential to transfer electrons extracellularly either to iron mineral particles or to microbial syntrophs, including methanogens. This is likely via transmembrane cytochromes, outer membrane hexaheme c-type cytochrome (OmcS) in particular, or pilin monomer, PilA, which were attributed to this lineage. Bonafide anaerobic oxidizers of methane (ANME) and denitrifying methanotrophs Methylomirabilia (NC10) may mediate AOM in these methanogenic sediments, however we also consider the role of methanogens in active AOM or back flux of methanogenesis. . Putative aerobes, such as methane-oxidizing bacteria *Methylomonas* and their methylotrophic synthrophs *Methylotenera,* are found among the anaerobic lineages in Lake Kinneret iron amended slurries and are also involved in the oxidation of methane or its intermediates, as suggested previously. We propose a reaction model for the metabolic interactions in these sediments, linking the potential players that interact via intricate metabolic tradeoffs and direct electron transfer between species. Our results highlight the metabolic complexity of microbial communities in an energy-limited environment, where aerobe and anaerobe communities may co-exist and facilitate AOM as one strategy for survival.

## 1. Introduction

Methane ($CH_4$) is an effective greenhouse gas (Wuebbles and Hayhoe 2002). The sources of methane to the atmosphere are of anthropogenic and natural origins, where natural methane is produced mainly biogenically (methanogenesis) as the final remineralization process in anoxic environments (Froelich et al. 1979). The natural sources of methane contribute up to 40% of the global methane emissions (Saunois et al. 2019), almost entirely from freshwater systems (Bastviken et al. 2011; Zhu et al. 2020). In these freshwater environments, anaerobic oxidation of methane (AOM) consumes over 50% of the produced

methane (Segarra et al. 2015). These environments are usually low in sulfate, and thus other terminal electron acceptors such as metals, iron (Fe) in particular, nitrate, nitrite and humic acids become available for this process, as explored in recent studies (Raghoebarsing et al. 2006; Ettwig et al. 2010; Adler et al. 2011; Haroon et al. 2013; Norði et al. 2013; Scheller et al. 2016; Bai et al. 2019). However, the diversity and metabolic potential of the microbial communities in natural anoxic ferruginous sediments are not fully understood (Vuillemin et al. 2018).

This study investigates the microbial communities in the sulfate-depleted, iron-rich methanic sediments of Lake Kinneret, a monomictic lake located in the north of Israel at mid-latitudes. The lake is thermally stratified from March until December. During the stratified period, due to the spring phytoplankton bloom decline, the hypolimnion of the lake becomes enriched with particulate organic carbon and oxygen is gradually depleted (Eckert and Conrad 2007). At the top 5 cm of the sediment,

sulfate reduction is the main process throughout the year (Eckert and Conrad 2007). At 5 cm below the sediment surface, methanogenesis becomes dominant (Adler et al. 2011). Methane production peaks at 7−12 cm in the sediment while sulfate is depleted (below 10 μM). At 20 cm depth, methane decrease is observed and ferrous iron concentrations increase to about 200 μM (Adler et al. 2011; Bar-Or et al. 2015). Iron-dependant AOM (Fe-AOM) appears to play a role in methane removal from this deep methanogenic zone of the lake sediments, based on porewater depth profiles, rate modeling, microbial

profiles and incubation experiments on top cores and slurries (Adler et al. 2011; Sivan et al. 2011; Bar-Or et al. 2015, 2017).

Complex microbial consortia mediate the biogeochemical transformations in anoxic lake sediments (Vuillemin et al. 2018). These include methylotrophic and hydrogenotrophic methanogens, from the Thermoplasmata, Methanomicrobia, Methanobacteria and Bathyarchaeota clades, as well as potential sulfate or iron reducers, such as Deltaproteobacteria,

Firmicutes and Nitrospirae lineages (Vuillemin et al. 2018). Alongside with the abovementioned lineages, Bar-Or et al. (2015) identified acetoclastic methanogens such as *Methanothrix* (formerly *Methanosaeta*, Methanosarcinales, recently reclassified as Methanothrichales by the Genome Taxonomy Database, GTDB, Parks et al., 2018), as well as the $H_2/CO_2$-using methanogenic Methanomicrobiales genera *Methanolinea* and *Methanoregula* in Lake Kineret sediments. Very few bona fide ANMEs were found, and the identity of microbes that perform Fe-AOM in these sediments was unknown.

Desulfuromondales (Deltaproteobacteria/Desulfobacterota) and Thermodesulfovibrio (Nitrospirae/Nitrospirota) were suggested to couple catabolism of organic substances to the reduction of metals (Bar-Or et al. 2015). We note that

Deltaproteobacterial lineages have been recently re-classified based on the GTDB (e.g. Desulfuromondales were re-classified as Desulfuromonadota phylum, Parks et al., 2018). Hereafter, we use the "Deltaproteobacteria" terminology to describe the taxonomy of these lineages, as implemented in the Silva132 database, as the new Genome Taxonomy Database

classification has not been peer-reviewed and widely accepted at the time of preparation of this manuscript.

Here, we use metagenomic analyses to explore the phylogenetic diversity and the metabolic potential of the microbial communities in the deep (27-41cm), iron-rich methanic part of Lake Kineret sediments (LK-2017) and the slurry incubations from the Bar-Or et al. 2017 study. These incubations, including a) $^{13}CH_4$, b) $^{13}CH_4$ + Hematite, or c) $^{13}CH_4$ + amorphous iron

+ molybdate (A.Fe(III)+MoO$_4$) produced substantial amounts of $^{13}$C-labelled dissolved inorganic carbon over 470 days (80-450‰, Fig. S1 in the Supplement). In all the treatments, the diversity of bacteria and archaea was similar to that of the natural sediments. We supplemented this data with analyses of bacterial and archaeal diversity at the 16S rRNA gene level for a wider range of treatments analyzed by Bar-Or et al. (2017). We sought genetic evidence for the ability of the microorganisms to perform iron reduction and produce or oxidize methane and aimed to identify and classify genes that are

necessary to catalyze reactions of the respective pathways. As in most natural sediments, iron and manganese are present mainly as low soluble oxide minerals at circumneutral pH (Norði et al. 2013; He et al. 2018) and microbial cells are impermeable to these solid minerals (Shi et al. 2007), we aimed to identify the strategies that may allow microorganisms to cope with this constraint and potentially perform metal-AOM, including (1) direct transfer of electrons to the mineral at the cell surface via electron carrier such as cytochrome c (Shi et al. 2007), (2) bridging electrons to a mineral via a cellular

appendage, such as conductive nanowires (Gralnick and Newman 2007; Schwarz et al. 2007; Shi et al. 2016; Lovley and Walker 2019), (3) indirect electron transfer via metal chelate (Paquete et al. 2014), and (4) indirect electron transfer by electron shuttling compounds, such as quinones or methanophenazines (Newman and Kolter 2000; Wang and Newman 2008; He et al. 2019).

## 2. Materials and methods

**2.1 Sample collection, DNA extraction, and sequencing:** Lake Kinneret sediments were collected in 2013 from the center of the lake (station A, located at 42 m depth), and slurry incubations were set up under anaerobic conditions, as previously described (Bar-Or et al. 2017). Briefly, sediment from the sediment zone of 26-41 cm was mixed with porewater extracted from parallel geochemical zone to create a homogenized 1:5 sediment to porewater ratio slurries. The homogenized slurry was transferred under continuous $N_2$ flushing in 40 mL portions into 60 mL bottles crimp and sealed. The sediment slurries

were amended with $^{13}$C-labeled methane and except the natural sample treated with different iron oxide minerals. Each sample set was kept (I) without inhibitors, (II) with inhibition of sulfate reduction and sulfur disproportionation by sodium molybdate (Na$_2$MoO$_4$) addition, and (III) inhibition of methanogenesis by 2-Bromo Ethane Sulfonate (BES) addition. One of the sample sets was killed (autoclaved) control. The various treatments are summarized in Supporting Material Table S1. DNA was extracted from each incubation at the beginning and end of the experiment (after 470 days) and analysis of DNA

16S rRNA genes was performed for all of these incubations and the untreated sediments (Supplementary Methods). Four metagenomic libraries (untreated sediments - t0-2013, incubations - $+^{13}CH_4$, $^{13}CH_4$+Hematite, and $^{13}CH_{4+}$Amorphous (A.) Fe(III)+MoO$_4$) were prepared at the sequencing core facility at the University of Illinois at Chicago using Nextera XT DNA library preparation kit (Illumina, USA). Our preliminary analyses of the microbial diversity (16S rRNA amplicons and metagenomics) in t0-2013 revealed contamination with common laboratory bacteria, such as Firmicutes and Bacilli (Figs. S2

a and S4 in the Supplement). To avoid the discovery of contaminant functions, we prepared a DNA library for an additional sample, collected from the same water and sediment depth in 2017 (LK-2017). 12-28 million $2 \times 150$ bp paired-end reads per library were sequenced using Illumina NextSeq500.

**2.2 Bioinformatics**: For each library, taxonomic diversity was determined by mapping the reads to Silva V132 database of

the small subunit rRNA sequences using phyloFlash (Glöckner et al. 2017; Gruber-Vodicka et al. 2019), or by using a protein-level classifier Kaiju (Menzel et al. 2016). The list of normalized taxa abundances, following removal of chloroplast and mitochondria sequences, was used as input for Fig. 1 and Fig. S4 in the Supplement. Metagenomes were co-assembled from concatenated reads from four metagenomic libraries (LK-2017, $+^{13}CH_4$, $^{13}CH_4$+Hematite, and $^{13}CH_4$+A.Fe(III)+MoO$_4$) with Spades V3.12 (Bankevich et al. 2012; Nurk et al. 2013), following decontamination, quality filtering (QV= 10) and

adapter-trimming with the BBDuk tool from the BBMap suite (Bushnell B, http://sourceforge.net/projects/bbmap/). Downstream analyses, including open reading frame (ORF) prediction, homology and hidden Markov models-based searches against taxonomic and functional databases, estimates of function abundance based on read coverage and automatic binning were performed with SqueezeMeta pipeline (Tamames and Puente-Sánchez 2019). ORFs and KEGG functions were quantified based on the mapping of metagenomics reads as counts per million (an equivalent of transcripts per million, TPM,

in transcriptomics). To predict the general metabolic functions, we assigned KEGG functions to the 21 categories of the Functional Ontology Assignments for Metagenomes (FOAM) database (Prestat et al. 2014). Automatic binning using metabat2 (Kang et al. 2015), maxbin (Wu et al. 2015), which was refined using DAStool (Sieber et al. 2018) and manual binning based on differential coverage and guanine-cytosine content with gbtools (Seah and Gruber-Vodicka 2015) resulted in a limited number of high-quality metagenome-assembled genomes, therefore in this study we looked for specific functions

in the metagenomes, followed by homology searches against the RefSeq (O'Leary et al. 2016) and GeneBank databases to evaluate taxonomy. Multiheme cytochromes (MHCs) were identified based on Cytochrome C Pfam HMM (PF00034) (Boyd et al. 2019), which revealed only a limited number of sequences. Thus, we identified open reading frames that comprised more than three cytochrome c binding motif sites (CxxCH) (Leu et al. 2020). Putative transmembrane (first 60 amino acids < 10, the expected number of amino acids in transmembrane helices > 18) and secreted peptides (first 60 amino acids ≥ 10)

were identified with TMHMM V2.0 (Moller et al. 2002). Putative transmembrane (first 60 amino acids < 10, the expected number of amino acids in transmembrane helices > 18) and secreted peptides (first 60 amino acids ≥ 10) were identified with TMHMM V2.0 (Moller et al. 2002). Secreted MHCs were also identified by SignalP v5.0, using the archaeal, Gram-negative and positive options (Almagro Armenteros et al. 2019), and the list was curated based on both TMHMM and SignalP

annotations. Other functions, including OmcS, PilA, HdrA, HdrD, HdrE, FwdC/FmdC (K00202), Ftr (K00672), Mch
(K01499), MtrA (K00577), Mer (K00320), Mtd (K00319), mcrA (K00399) and Fpo subunits (K22158-K22170) were
identified based on their KEGG orthology, and their taxonomy was assigned based on BLAST searches versus RefSeq
database (amino acid sequences are provided as supplementary files S.DB.6-8). For conservative predictions of electroactive
pilins, we selected PilA sequences with ≥9.8% aromatic amino acids and ≤22-aa aromatic-free gap using a python script
([https://github.com/GlassLabGT/Pythonscript](https://github.com/GlassLabGT/Pythonscript)), as previously described  (Bray et al. 2020).

## 3. Results and Discussion

### 3.1 Geochemical evidence for iron coupled AOM in Lake Kinneret iron-rich methanic sediments

We explore here slurries amended with Lake Kinneret sediments from the deep methanic zone (26-41 cm). In this potentially
ferruginous zone, sedimentary profiles show that the concentration of methane decreased from its maximum values of above
2mM at around 10 cm depth to 500 µM at 40 cm depth, and that of dissolved ferrous iron increased (from 1-6µM at the first
10 cm below the sediment surface to ~60-100µM, depending on sampling season). This, combined with an increase of $\delta^{13}C$
of methane (from about -65‰ at 7 cm to -53‰ at 24 cm below the sediment surface) and a decrease of $\delta^{13}C$ of total lipid
compounds (from 27‰ at 23 cm to -31‰ at 27 cm below the sediment surface), suggest AOM in the deep sediment coupled
to iron reduction (Adler et al. 2011; Sivan et al. 2011). The iron coupled AOM process was supported by rate modeling and
by microbial profiles (Adler et al. 2011; Sivan et al. 2011; Bar-Or et al. 2015, 2017). We note that alternative common
electron acceptors were scarce: dissolved manganese oxides concentrations were ~ 0.04% and nitrate and sulfate were below
the detection limit (Sivan et al. 2011).

We amended the slurries with isotopically labeled $^{13}CH_4$, $^{13}CH_4$ + hematite and $^{13}CH_4$ + amorphous iron + molybdate for 470
days. Methane concentration increased with time (by up to ~650µM in the treatment with hematite addition, up to ~1000µM
enrichment in the natural treatment and up to ~300µM in the treatment with amorphous iron + molybdate), reflecting net
methanogenesis.   In these incubations, we observed also a marked enrichment of labeled carbon in the DIC pool after ten
months of incubation (up to 250‰ enrichment in the treatment with hematite addition, up to 80‰ enrichment in the natural
treatment and up to 450‰ in the treatment with amorphous iron + molybdate, Fig. S1 in the Supplement), indicating
conversion of methane to DIC. Ferrous iron concentrations increased by ∼20−50 µM following iron oxide amendments
(with and without molybdate addition), indicating that iron was reduced. The BES amendments resulted in the highest
increase in ferrous iron concentrations (~50-110 µM), most likely due to the abiotic reaction of BES with iron minerals. The
evidence for iron reduction, together with the fact that $\delta^{13}C_{DIC}$ values increased by 250-450‰ in the different iron amended
treatments, but not in methane-only additions (only up to 80‰, Fig. S1 in the Supplement), suggest iron coupled AOM.
Sulfate did not seem to play a role in the AOM, as the addition of molybdate, sulfate reduction and disproportionation
antagonist, did not inhibit methane turnover (Fig. S1 in the Supplement). The addition BES to specific slurries inhibited the
production of $\delta^{13}C_{DIC}$, indicating the essential role of methanogens in the AOM activity (Fig. S1 in the Supplement).

**3.2 Diverse microbial consortia mediate biogeochemical cycles in Lake Kinneret sediments**

Diverse microbial consortia inhabit Lake Kinneret iron-rich methanic sediments (>20 cm depth, Fig.1). In these sediments, Bacteria outnumber Archaea based on mapping of the metagenomic reads either to the Silva (V132) database of the 16S rRNA gene sequences (73-76% and 24-27% reads mapped to bacterial and archaeal sequences, respectively, Supplementary Dataset.1) or to MAR (MARine) database of prokaryotic genomes (81-85% and 15-19% reads mapped to bacterial and archaeal sequences, respectively, Supplementary Dataset. 2). We explored here mainly the microbial community in the deep sediment incubations (>20 cm) amended with $^{13}CH_4$ alone, or $^{13}CH_4$ with hematite, or $^{13}CH_4$ with amorphous iron oxides plus molybdate using metagenome analyses (Fig.1). This diversity of microbes resembled that described previously for Lake Kinneret sedimentary profiles with amplicon sequencing of the 16S rRNA gene (Bar-Or et al. 2015), as well as that determined with either amplicon sequencing or metagenomics in ferruginous lakes across the globe (e.g. Vuillemin et al. 2018; Kadnikov et al. 2019). According to the metagenome analyses, the variation in the diversity of microbial communities in the natural samples and the incubations was small, possibly because iron is not a limiting electron acceptor throughout the sampling interval in the sediments (Sivan et al. 2011). Amplicon sequencing of the bacterial and archaeal 16S rRNA genes in a wider range of treatments (Figs. S2 and S3 in the Supplement), which included additions of various iron minerals, as well as amendments of molybdate (inhibitor of sulfate reduction) and BES (inhibitor of methanogenesis), also revealed minor changes in the phylogenetic diversity of the microbial consortia.

Anaerolineales (Chloroflexi), Thermodesulfobrionia (Nitrospirae) and the deltaproteobacterial Sva0485 clade were among the most dominant bacterial lineages in these samples (3-6% read abundance, Fig.1). While all of these lineages may carry out dissimilatory sulfate reduction (Vuillemin et al. 2018), genetic evidence suggests that bacteria from the Sva0485 clade, which was recently named *Candidatus* Acidulodesulfobacterales, have the potential to reduce iron (Tan et al. 2019). Sva0485 was suggested recently to be involved in iron reduction also in the methanic zone of marine sediments (Vigderovich et al. 2019). Both these studies revealed a correlation between the distribution of this lineage and ferrous iron concentrations in sediment porewater. Interestingly, *Ca.* Acidulodesulfobacterales appear to thrive mainly under acidic conditions, while Thermodesulfobrionia was suggested to be either neutrophilic or alkaliphilic (Sekiguchi et al. 2008; Frank et al. 2016). Thus, the co-occurrence of these taxa hints that microenvironments with distinct physicochemical conditions may be present in Lake Kinneret sediments.

Methanomicrobiales (5-7% reads abundance) and Bathyarchaeia (4-7% read abundance) were the most dominant archaeal lineages in the sediment (Fig.1). The vast majority of the Methanomicrobiales sequences belonged to *Methanoregulaceae* genera *Methanoregula* and *Methanolinea*, which are known to obtain energy by $CO_2$ reduction to methane, with $H_2$ or

formate as electron donors (Bräuer et al. 2015; Imachi and Sakai 2016). Bathyarchaeia may occur in large numbers in lake sediments (e.g. Vuillemin et al. 2018; Kadnikov et al. 2019; Zhang et al. 2019). The role of these generalists, which are capable of using various carbon and energy sources, often fueling their metabolism through acetogenesis and methanogenesis, is not well understood (Evans et al. 2015; He et al. 2016; Yu et al. 2018; Zhou et al. 2018). Other notable archaeal lineages included the acetoclastic *Methanothrix* (1-3% read abundance), which are often found in anoxic lake sediments (Smith and Ingram-Smith 2007; Schwarz et al. 2007; Carr et al. 2018) as well as in oxygenated soil, as recently discovered for *Methanothrix paradoxum* (Angle et al. 2017). Their role in the anaerobic degradation of alkanes as syntrophs of *Anaerolineaceae* spp. has been recently proposed based on enrichment culturing (Liang et al. 2015), as well as their ability to oxidize methane (Valenzuela et al., 2017, 2019). We also identified the putative obligate $H_2$-dependent methylotrophic methanogen lineages Methanofastidiosales (Thermococci, 1-3% read abundance) and Methanomassiliicoccales (Thermoplasmata 0.4-1% read abundance), as well as the putative degraders of detrital proteins Marine Benthic Group D (Thermoplasmata, 1-2% read abundance) (Lazar et al. 2017; Evans et al. 2019). Less than 1% of the total reads were mapped to sequences of the anaerobic methanotrophs, such as ANME-1 (0.3-0.8%), as well as those of the nitrite-reducing methane oxidizers Methylomirabilales (NC10, 0.3-0.6%, Supplementary Dataset. 1).

Some type I Methylococcales methanotrophs were found (0.4-1.8%) in Lake Kineret sediments. This finding is supported by the quantitative polymerase chain reaction (qPCR) analyses of the *pmoA* gene (Bar-Or et al. 2017), our analyses of bacteria diversity at the 16S rRNA gene level (Fig. S2 in the supplement) and the [13]C-labelled methane carbon incorporation in phospholipid-derived fatty acids that are typical of type I methanotrophs (Bar-Or et al., 2017), suggesting that methane metabolism was active in these bacteria. Methylotrophic *Methylotenera* (recently reclassified as Burkholderiales, Betaproteobacteriales in Silva132), which were shown to co-occur with type I methanotrophs under nearly hypoxic conditions (Beck et al. 2013; Cao et al. 2019), were also found (0-1%, Supplementary Dataset.1). The mechanisms behind the increase and elevated activity of the presumingly aerobic methanotrophs in the anoxic sediments have not been fully understood, although this phenomenon appears to be widespread (Bar-Or et al. 2017; Martinez-Cruz et al. 2018).

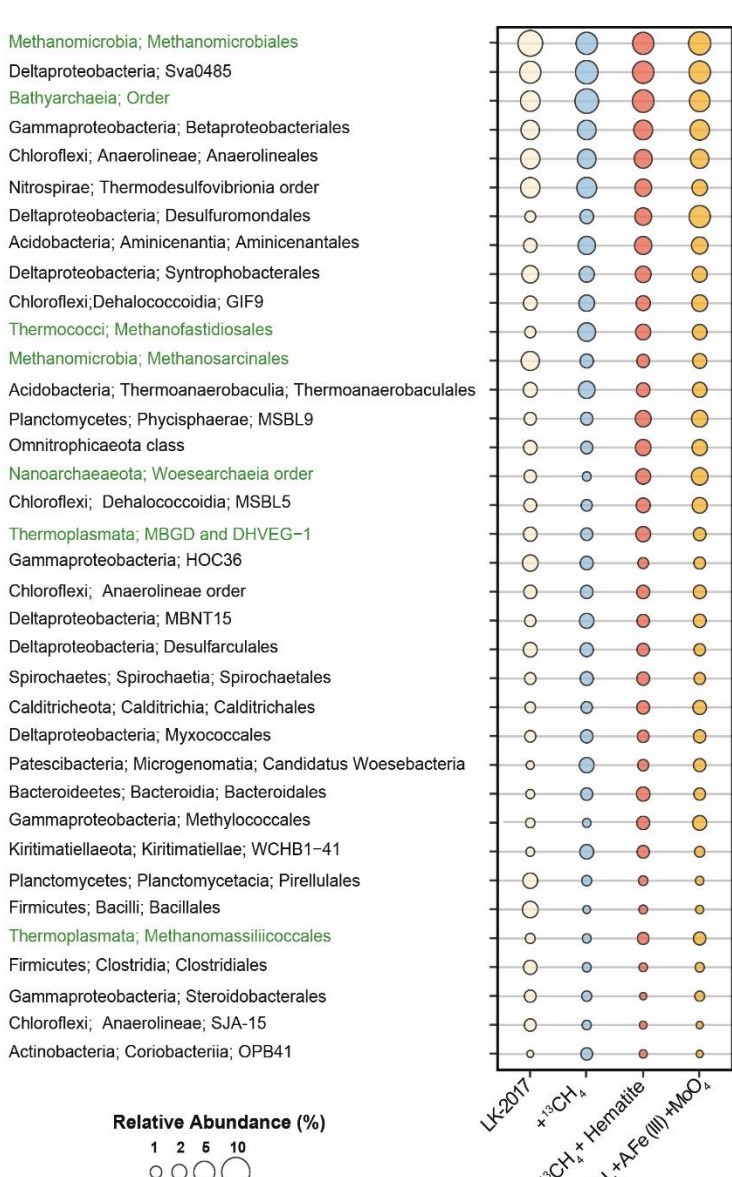

Methanomicrobia; Methanomicrobiales
Deltaproteobacteria; Sva0485
Bathyarchaeia; Order
Gammaproteobacteria; Betaproteobacteriales
Chloroflexi; Anaerolineae; Anaerolineales
Nitrospirae; Thermodesulfovibrionia order
Deltaproteobacteria; Desulfuromondales
Acidobacteria; Aminicenantia; Aminicenantales
Deltaproteobacteria; Syntrophobacterales
Chloroflexi;Dehalococcoidia; GIF9
Thermococci; Methanofastidiosales
Methanomicrobia; Methanosarcinales
Acidobacteria; Thermoanaerobaculia; Thermoanaerobaculales
Planctomycetes; Phycisphaerae; MSBL9
Omnitrophicaeota class
Nanoarchaeaeota; Woesearchaeia order
Chloroflexi; Dehalococcoidia; MSBL5
Thermoplasmata; MBGD and DHVEG−1
Gammaproteobacteria; HOC36
Chloroflexi; Anaerolineae order
Deltaproteobacteria; MBNT15
Deltaproteobacteria; Desulfarculales
Spirochaetes; Spirochaetia; Spirochaetales
Calditricheota; Calditrichia; Calditrichales
Deltaproteobacteria; Myxococcales
Patescibacteria; Microgenomatia; Candidatus Woesebacteria
Bacteroideetes; Bacteroidia; Bacteroidales
Gammaproteobacteria; Methylococcales
Kiritimatiellaeota; Kiritimatiellae; WCHB1−41
Planctomycetes; Planctomycetacia; Pirellulales
Firmicutes; Bacilli; Bacillales
Thermoplasmata; Methanomassiliicoccales
Firmicutes; Clostridia; Clostridiales
Gammaproteobacteria; Steroidobacterales
Chloroflexi; Anaerolineae; SJA−15
Actinobacteria; Coriobacteriia; OPB41

Relative Abundance (%)
1  2  5  10

LK-2017
+¹³CH₄
¹³CH₄ + Hematite
¹³CH₄ +AFe (III) +MoO₄

Fig. 1. Relative abundance of Bacteria (black) and Archaea (green) at the order level based on mapping of metagenomic reads to Silva132 database of the small subunit rRNA sequences. Lineages <1%, which account together for 28-32% of the microbial community, were removed from the display. LK 2017 refers to the natural sediments at depth 26 cm at t0

**3.3 The general metabolic potential of microbial communities**

General metabolic potential:

Based on the mapping of the metagenomics reads to open reading frames (ORFs), for which KEGG orthology was assigned, several metabolic pathways were found to be dominant in Lake Kinneret iron-rich methanic sediments (Fig. 2). Overall, all four samples had a similar metabolic repertoire. The vast majority of mapped reads were attributed to amino acid utilization and biosynthesis, suggesting that the turnover of organic nitrogen plays an important role in fueling these microbial communities. Indeed, the ORFs that encode the five components of the branched-chain amino acid transport system (KEGG

IDs KO1995-KO1999) were among the top 30 most abundant KEGG functions (out of a total of 9058 KEGG IDs that were assigned to the metagenomics ORFs, Supplementary Dataset.3). Circa 97% of ORFs that encoded these components were assigned to bacteria (mainly Deltaproteobacteria and Chloroflexi, 46% and 6% of reads mapped). The main intermediates of amino acid catabolism are fatty acids (Park et al. 2014; Narihiro et al. 2016; Aepfler et al. 2019; Scully and Orlygsson 2019). Indeed, the KEGG IDs that are associated with the β-oxidation of fatty acids, including long-chain acyl-CoA synthetase (K01897), aldehyde:ferredoxin oxidoreductase (K03738) and acetyl-CoA C-acetyltransferase (K00626) were among the top 10 most abundant functions, based on read mapping (acyl-CoA dehydrogenase K00249 was among the top 50 most abundant functions, Supplementary Dataset.3). Under anoxic conditions, the catabolism of many amino acids and their fatty acid derivatives is not thermodynamically favorable, thus this process is often coupled to syntrophic hydrogenotrophic methanogenesis or sulfate reduction (Sieber et al. 2012; Scully and Orlygsson 2019; Ziels et al. 2019). In Lake Kinneret deep sediments (~>20 cm), the abundant hydrogenotrophic Methanomicrobiales methanogens are the likely syntrophic scavengers of hydrogen. The hydrogen concentration in the deep methanic horizon is ~20 µM gr$^{-1}$ sediment (Adler 2015). Given that sulfate is below the detection limit there (<10µM, Adler et al., 2011, Sivan et al., 2011), hydrogen scavenging may also be coupled to metal reduction, most likely by Deltaproteobacterial lineages, some of which may be syntrophic (e.g. Syntrophobacterales). Through syntrophy, amino acids can be converted to acetate and propionate (Narihiro et al. 2016), which further fuel other organisms, including among others, acetoclastic methanogens.

Notably, the most well-represented KEGG function was the heterodisulfide reductase subunit A (HdrA, K03388, Supplementary Dataset.3). Heterodisulfide reductases play an important role in the energy metabolism of anaerobic bacteria and archaea (such as sulfate reducers and methanogens), driving flavin-based electron bifurcation from various electron donors (Pereira et al. 2011; Ramos et al. 2015; Wagner et al. 2017; Buckel and Thauer 2018). The HdrA protein itself carries the flavin adenine dinucleotide that is needed for bifurcation (Wagner et al. 2017). In our samples, read mapping suggests that the major taxonomic groups that carry the *hdrA* genes include Deltaproteobacteria (13-26%), Chloroflexi (14-21%), Bathyarchaeia (6-10%) and Methanomicrobiales (3-4%) as well as other bacteria and archaea (grouped under "below 3%", Fig. S5 in the supplement). These results highlight the presence of bifurcation-driven metabolic processes in Lake Kinneret sediments.

In agreement with previous metagenomic assessments of metabolic pathways in microbes from anoxic lake sediments (Vuillemin et al. 2018), fermentation and methanogenesis account for a substantial part of the metabolic repertoire (Fig. 2a). KEGG IDs that are associated with the fermentative metabolism, such as formate dehydrogenase (K00123), 2-oxoglutarate ferredoxin oxidoreductase (K00174-5), acetolactate synthase (K01652) were among the functions with the highest metagenomic coverage (Supplementary Dataset.3). Functional analysis of the metagenome suggests that carbon dioxide, formate, acetate, and methylated compounds can fuel methanogenesis (Fig. 2b). These findings are in line with the fact that archaeal lineages known to be capable of using the respective pathways were present (Fig. 1).

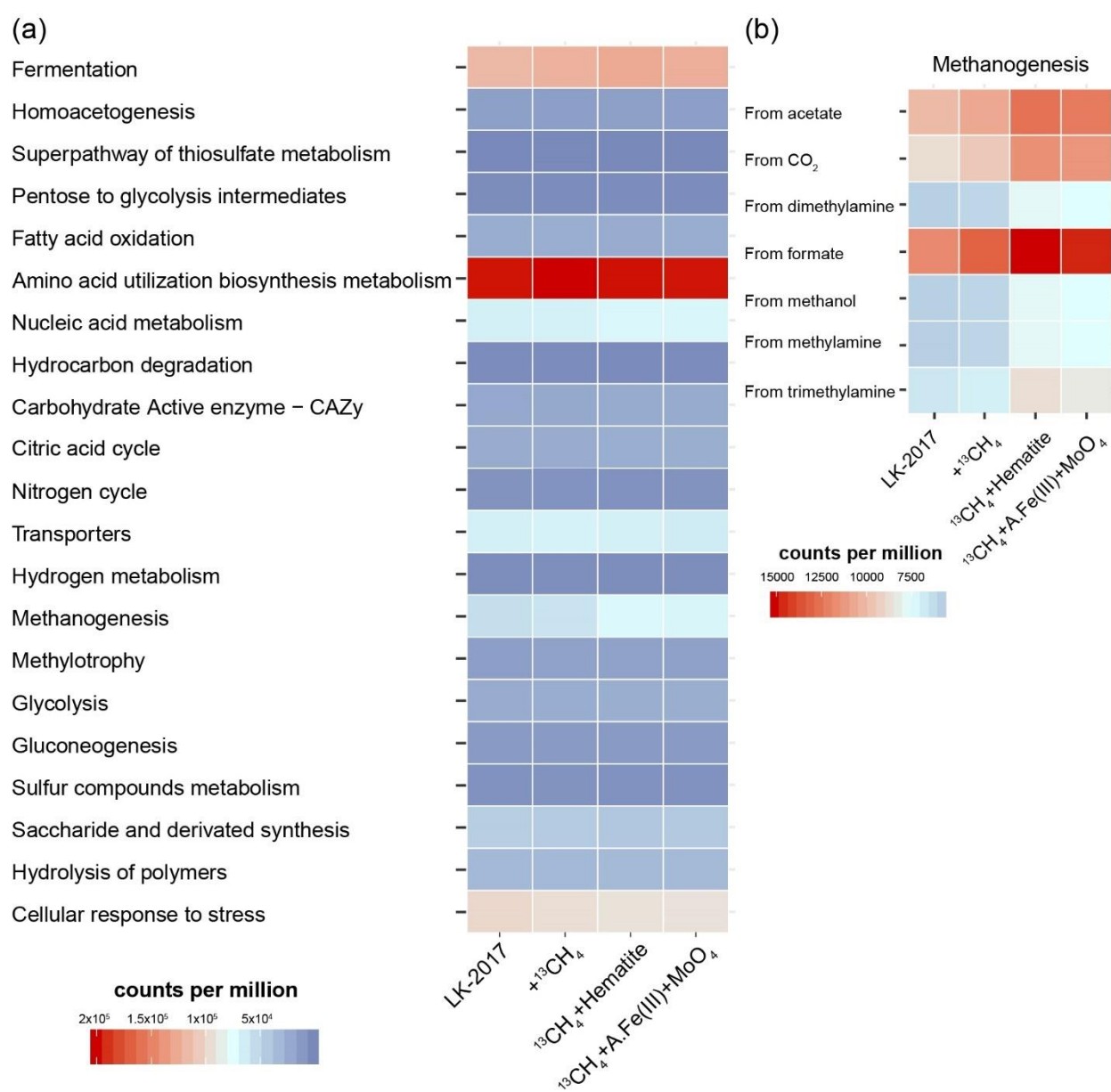

Fig.2. The metabolic potential of microbial communities in Lake Kinneret sediments and slurry incubations. Reads were mapped to metagenomic ORF for whichKEGG functions were assigned. KEGG functions were organized and grouped into FOAM (Functional Ontology Assignments for Metagenomes) categories. The relative abundance of each FOAM category is presented in counts per million (CPM). (a) General pathways (b) Zooming into the pathway of methanogenesis. A.Fe(III)+MoO$_4$= amorphous iron and molybdate.

Metabolic potential for AOM:

In the incubations, methane oxidation was indicated based on the transformation of labeled $^{13}$C-methane to $^{13}$C-DIC and was
suggested to involve methanogens due to the inhibition of this transformation by BES (inhibitor of the methyl coenzyme M
reductase). Based on the taxonomic assignment of genes that encode the enzymes in anaerobic methane metabolism
pathways, archaeal lineages from Lake Kinneret iron-rich methanic sediments, such as Methanomicrobiales,
Methanosarcinales, and Methanomassiliicoccales (66%, 26% and 4% of reads mapped to K00399 McrA-encoding ORFs,
respectively), are capable of performing the methane transformations (Fig. 3). As described above, all of these lineages are
methanogens that convert inorganic carbon, acetate, and methylated compounds to methane under most environmental
conditions. No *mcrA* sequences were assigned to Bathyarchaeia (Fig. 3), in agreement with previous observations of the rare
occurrence of the *mcrA* gene in this lineage (Evans et al. 2015; Lazar et al. 2015; He et al. 2016; Maus et al. 2018). Thus,
Bathyarchaeia is most likely not involved directly in methanogenesis or methanotrophy.

AOM by reversal of methanogenesis is unlikely in obligate hydrogenotrophic methanogens Methanomicrobiales (Rotaru et
al. 2014) and it has not been shown also in methylotrophic the methanogens *Candidatus* Methanofastidiosa and
Methanomassiliicoccales (Lang et al. 2015; Nobu et al. 2016; Yan and Ferry 2018). More likely candidates to perform this
AOM in Lake Kinneret sediments are ANME-1, which has been shown to perform this process and exist in our deep
sediment, and *Methanothrix*. *Methanothrix* is closely related to Methanosarcinales (now classified as order
Methanothrichales in class Methanosarcinia)  and was suggested to perform AOM (Valenzuela et al., 2017, 2019). Several
studies suggest that reverse methanogenesis may occur in Methanosarcinales species, based on (1) observation of trace
methane oxidation (Zehnder and Brock 1979; Moran et al. 2005), (2) catalyzation of reverse methanogenesis upon insertion
of an *mcr* gene clone of ANME-1 (Soo et al. 2016), and (3) phylogenetic affiliation of bonafide ANMEs such
Methanoperedenaceae (ANME-2d) and ANME-2a with this clade (Cai et al. 2018; Yan et al. 2018). *Methanothrix* was
shown to receive electrons from *Geobacter* or other organotrophs such as *Sphaerochaeta* during acetoclastic growth (Rotaru
et al. 2014; Zhao et al. 2020), suggesting that its cell surface may be conductive. Moreover, the electron transfer enables this
strict acetoclastic methanogen to use these electrons and reduce $CO_2$ to methane, a metabolic capability that was unknown in
these organisms previously. In line with the abovementioned study, our metagenomics analyses revealed that all the seven
enzymes needed to perform both forward and reverse methanogenesis (Fmd/Fwd, Ftr, Mch, Mtd, Mer, Mtr and Mcr) were
assigned to Methanosarcinales (Fig. 3). We note that since we measured net methanogenesis, the observed $^{13}$C-DIC
enrichment can represent the back flux of the close-to-equilibrium methanogenesis. In principle, any methanogen can
produce DIC via the back‑flux reactions, yet we cannot estimate the isotopic fingerprint and the magnitude of this process in
this set of incubations.  Thermodynamically, both scenarios (back flux or active Fe-AOM) are plausible, and the bona fide

Fe-AOM is feasible and competitive with organoclastic iron reduction (ΔG of about -200 kJ/mol for both processes (Table S2 in the supplement).

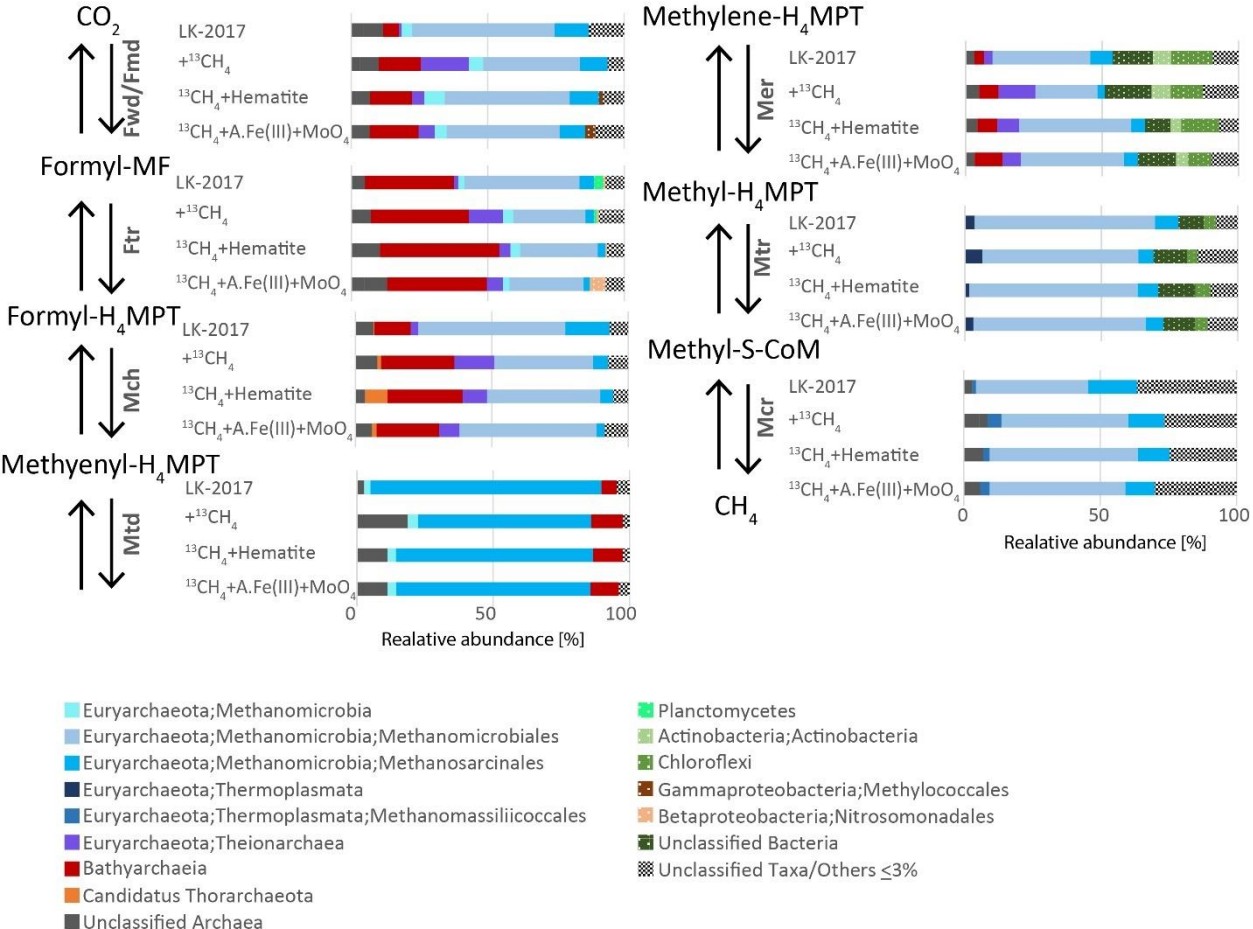

Fig.3. Phylogenetic diversity of the seven core genes of methanogenesis from $CO_2$. Phylogenetic assignments are based on BLAST mapping against the RefSeq database. Taxonomic classifications at the highest level possible (up to the order level) are shown. A.Fe(III)+$MoO_4$ = amorphous iron and molybdate.

## 3.4 Genomic evidence for the microbial iron reduction in Lake Kinneret sediment

We asked whether direct extracellular electron transfer, either to a mineral or interspecies, could potentially play a role in Lake Kinneret methane oxidation, in case it is active Fe-AOM, and which taxa may be involved in this process. Direct interspecies electron transfer (DIET) between electrogenic microbes, such as Desulfuromondales, and their partners, such as Methanosarcinales methanogens, requires the presence of electrically conductive pili (e-pili) on the deltaproteobacterial partner, while in both DIET and Fe (metal)-AOM the archaeal partner/methane oxidizer needs to produce either secreted or membranal conductive entities (Rotaru et al. 2014; McGlynn et al. 2015; Holmes et al. 2018; Yan et al. 2018; Walker et al.

2020; Leu et al. 2020). It was previously suggested that thermophilic AOM coupled to sulfate reduction is conducted via DIET between ANME-1 and sulfate-reducing bacteria (Wegener et al. 2015), thus we examined if DIET might also contribute to Fe-AOM in Lake Kinneret.

Our metagenomics results suggest that Lake Kinneret microbiota may transfer electrons directly to extracellular minerals, either through multiheme c-type cytochromes (MHCs) or microbial nanowires, according to mechanisms described in previous reviews (Lovley 2011; Shi et al. 2016). We investigated the trans-membranal MHCs that are anchored either in the bacterial membrane or archaeal S-layer, as well as secreted MHCs. The putative transmembrane and secreted MHCs were encoded by 66 and 592 ORFs per metagenome, respectively. In both MHC types, most of the ORFs were classified as
Deltaproteobacterial (Fig.4, 36-52% in transmembrane MHC and 29-35% in secreted MHCs). This is not surprising, as Deltaproteobacterial lineages are known to reduce particulate metals through extracellular electron transfer (EET), as well as to conduct DIET (Leang et al. 2003; Reguera et al. 2005; Lovley 2011; Adhikari et al. 2016). Other MHC sequences were associated with Nitrospirae, Chloroflexi, and Acidobacteria (both secreted and trans-membranal) and with Actinobacteria (secreted MHC , Fig. 4). Very few or none archaeal sequences were detected in both types of MHC (0-1.5%).

In Deltaproteobacteria (Desulfuromonadota) such as *Geobacter* and *Synthrophus*, the protein nanowires are assembled from PilA monomers (Lovley and Walker 2019; Walker et al. 2020). The role of C-type cytochrome OmcS nanowires in DIET has been recently suggested and debated (Filman et al. 2019; Wang et al. 2019; Lovley and Walker 2019). We surveyed the metagenome for the presence of both PilA and OmcS-encoding ORFs, most of which were indeed classified as
deltaproteobacterial sequences by BLAST against the NCBI nr/nt database (Fig. 4 c,d). The overall abundance of the MHC (secreted and trans-membranal), PilA and OmcS ORFs was 364-493, 35-45, 5-9 and 4-9 counts per million reads mapped, respectively. Our findings confirm that the phylogenetic diversity of microbes are capable of nanowire-mediated DIET extends beyond deltaproteobacterial lineages (Bray et al. 2020), as strict searches attributed pilA-like sequences not only to Desulfobacterota (Deltaproteobacteria), but also to Thermodesulfovibrionales, Burkholderiales, Gemmatimonadales,
Aminicenantales, as well as WOR-3 and Firmicutes (Fig, 4d). The identified OmcS ORFs were predominantly classified as deltaproteobacterial (36-55%), and some were assigned as Actinobacteria (0-23%, Fig. 4c). The vast majority of classified deltaproteobacterial OmcS hits were assigned as Desulfuromondales at the order level (represent 36-55% overall and 66-100% out of deltaproteobacteria). The evidence for the presence of bacteria that transfer electrons via nanowires not only implies DIET, but also indicates the potential of sediment microbiota to conduct EET using the particulate metals (Lovley
2011; Liu et al. 2019b), and thus mediate iron and manganese reduction.

Some methanogens, such as *Methanosarcina barkeri* and *Methanothrix* sp.*,* do not possess outer-surface MHCs, yet they are capable of DIET-based syntrophy (Rotaru et al. 2014; Holmes et al. 2018; Yee and Rotaru 2020). Tubular sheaths that are made of major sheath protein (MspA) may enable some related archaea, including *Methanothrix thermophila* and

*Methanospirillum hungatei*, to conduct electron transfer (Dueholm et al. 2015; Christensen et al. 2018; Liu et al. 2019a). The
protein sheet of *Methanothrix shoegheni*i was described to concentrate metal ions like iron, copper, nickel, and zinc (Patel et
al. 1986). It was thus proposed to give them an advantage in retrieving electrons from extracellular mineral-rich
environments (Yee and Rotaru 2020), as our study site. BLAST searches using *M. thermophila* and *M. hungatei* MspA
queries (ABK14853.1 and WP_011449234.1) in Lake Kinneret metagenomes resulted only in poor (11.9-27.5%) hits of the
WP_011449234.1 query, all of which were annotated as a hypothetical Methanoregulaceae protein. Thus, no known
Methanosarcinales MspA proteins were detected, and it is still unclear if and how these archaea conduct DIET.

Of the ANME playing a role in AOM, ANME-2d was recently proposed to perform MHC-mediated metal-AOM (McGlynn
et al. 2015; Ettwig et al. 2016; Fu et al. 2016; Scheller et al. 2016; Cai et al. 2018). Thus, we used the previously published
ANME-2d MHC sequences (Supplementary Dataset. 4) as a query for BLASTing against the metagenomes. This analysis
resulted in zero MHC BLAST hits, corresponding to the fact that only ANME-1 were found among ANME (as 16S rRNA
gene sequences and low-quality bins), while ANME-2d were absent. Other Methanosarcinales lineages such
as *Methanosarcina acetivorans,* were also suggested to conduct metal-dependent AOM (Cai et al. 2018; Yan et al. 2018; Leu
et al. 2020). In Methanosarcinales, the dimeric membrane-bound HdrDE complex catalyzes the oxidation of CoM and CoB
to CoMS-SCoB and electron shuttling within the membranes is mediated by methanophenazines, which appear to be
important in the reduction of extracellular iron by this lineage (Bond and Lovley 2002; Sivan et al. 2016; Bar-Or et al. 2017;
Yan et al. 2018; Holmes et al. 2019). While the taxonomic assignments of ORFs that encoded the HdrD subunit (K08264)
were diverse (34-39% Deltaproteobacteria, 9-14% Chloroflexi, 2-4% Bathyarchaeia and 1.5-3% Methanosarcinales, Fig. S5
in the supplement), 77-97% of all the ORFs that encoded the HdrE subunit (K08265), belonged to Methanosarcinales genus
*Methanothrix* (Fig. S5 in the supplement). However, these sequences were scarce (3:1000 *hdrE* to *hdrA* based on read
mapping). We also detected all the possible genes involved in the formation of the membrane-bound coenzyme
$F_{420}$:methanophenazine dehydrogenase complex Fpo (*fpoABCDHIJKLMNO*), which couple reduction of $F_{420}H_2$ with
methnophenazine oxidation and proton translocation (Welte and Deppenmeier 2014; Evans et al. 2019; Holmes et al. 2019),
and the majority of Fpo-encoding ORFs were classified as *Methanothrix* (Supplementary Database. 5). As we were unable to
identify Methanosarcinales MHCs or proteins that encode other conductive features such as the tubular sheaths, the question
regarding their involvement in Lake Kinneret metal-AOM remains open.

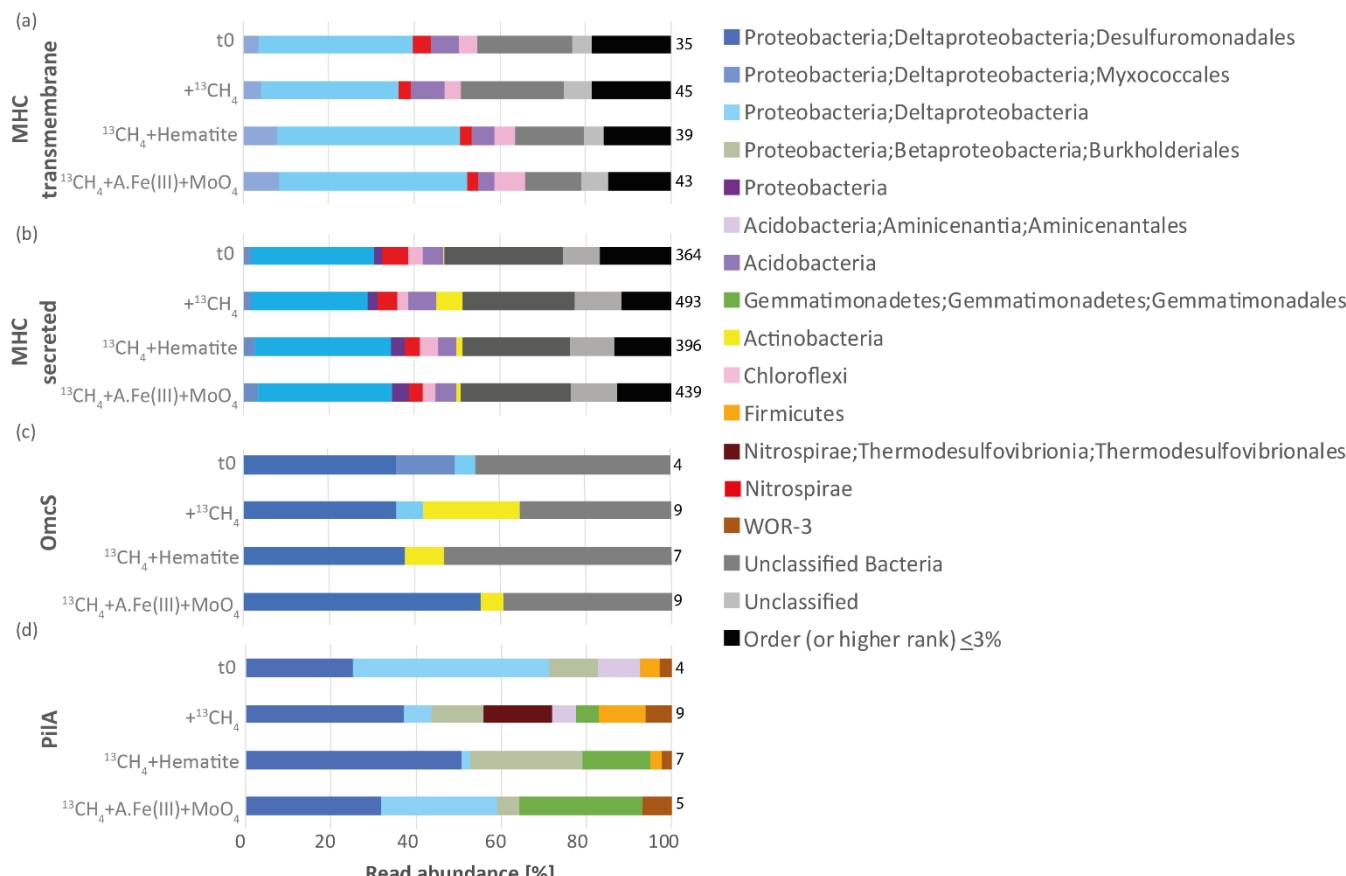

Fig.4. Taxonomic affiliation of open reading frames that are needed for extracellular electron transfer, and their relative abundances based on the read mapping. (a) transmembrane multiheme c-type cytochromes (MHC), (b) secreted MHC (c) outer membrane hexaheme c-type cytochrome (OmcS) and (d) fimbrial protein PilA. Numbers of total reads per million reads mapped to a gene according to the treatments are listed on the right side of each bar-plot. A.Fe(III)+MoO$_4$= amorphous iron and molybdate. We note that the taxonomy of several lineages here is based on the NCBI classification and differs from that of the Genome Taxonomy Database (GTDB). In particular, Deltaproteobacteria have been reclassified in the GTDB, thus the Desulforomandales lineages described here belong to the phylum Desulfobacterota, class Desulfuromonadia and Myxococcales to phylum Myxococcota. The phylum Nitrospirae is reassigned as Nitrospirota, Betaproteobacteria as order Burkholderiales in Gammaproteobacteria.

## 4. Summary

Metagenomic analyses of natural deep iron-rich sediments of Lake Kinneret and slurry incubations from this zone suggest that similarly to other freshwater systems (e.g. Vuillemin et al. 2018), a community of bacteria and archaea drives the mineralization of organic matter in the deep part of Lake Kinneret sediments, through degradation of amino and fatty acids, as well as hydrogenotrophic, acetoclastic and methylotrophic methanogenesis (Fig. 5).

We present here a reaction model of possible methane and iron cycling routes (Fig.5). Metagenomics suggests that the fermenters of amino acids and other products of necromass degradation are the abundant Anaerolineaceae, Thermodesulfobrionia, SVA0485 and Bathyarchaeia. One of the major end products of fermentation is acetate, which can be

used as a substrate for the acetoclastic methanogens. Both the acetoclastic methanogenesis and $CO_2$ reduction to methane in these species are likely driven by syntrophy with Desulfuromondales spp., through DIET (Rotaru et al. 2014; Inaba et al. 2019; Wang et al. 2020). The syntrophy between Methanosarcinales with Desulfuromondales is likely in Lake Kinneret deep sediments, based on the fact that the vast majority of ORFs that are needed for the extracellular transfer of electrons were assigned to Desulfuromondales.

Our geochemical isotope labeling experiments suggest that AOM is plausible in the deep methanogenic sediments. AOM may be mediated by ANME, such as ANME-1 that were scarce, may be coupled to iron reduction and may involve methanogens which have not been considered as bonafide ANME. (Fig. 5). However, we cannot rule out the role of methanogenesis back flux reactions, which could have produced the isotopically labeled DIC. Both modes of methane oxidation, via the Fe-AOM by an unknown methanogen and the methanogenesis back flux, are justified from the thermodynamic and kinetic perspectives.

Our results, as well as the previous analyses of fatty acids (Bar-Or et al. 2017), suggest that the aerobic methane-oxidizing *Methylomonas* and its aerobic methylotrophic partner *Methylotenera* also have a role in methane oxidation in the anaerobic environment. However, the mechanism behind this process is unclear. One possibility is that a slow release of oxygen from particulate matter (Wang et al. 2018) could have fueled methane oxidation by Methylococcales, given that oxidation of methane in the absence of oxygen is unlikely. The alternative is that these lineages may be able to incorporate methanol derived from the incomplete process of reverse methanogenesis (Fig. 5), as was shown for ANMEs (Xin et al. 2004; Wegener et al. 2016). Another possible explanation for the methylated compound leakage is the reversibility of the enzymes involved in AOM (Thauer and Shima 2008; Holler et al. 2011), which may lead to the formation of trace amounts of methylated substrates (Wegener et al. 2016). This option of methanol as an intermediate produced by the methanogenic archaea fits with the reported inhibition of methane oxidation upon the addition of BES (Bar-Or et al. 2017). This, as well as the functions of numerous other lineages that comprise the diverse consortia of Lake Kinneret deep iron-rich sediments, remain to be elucidated through further sequencing efforts, cultivation and experimental studies.

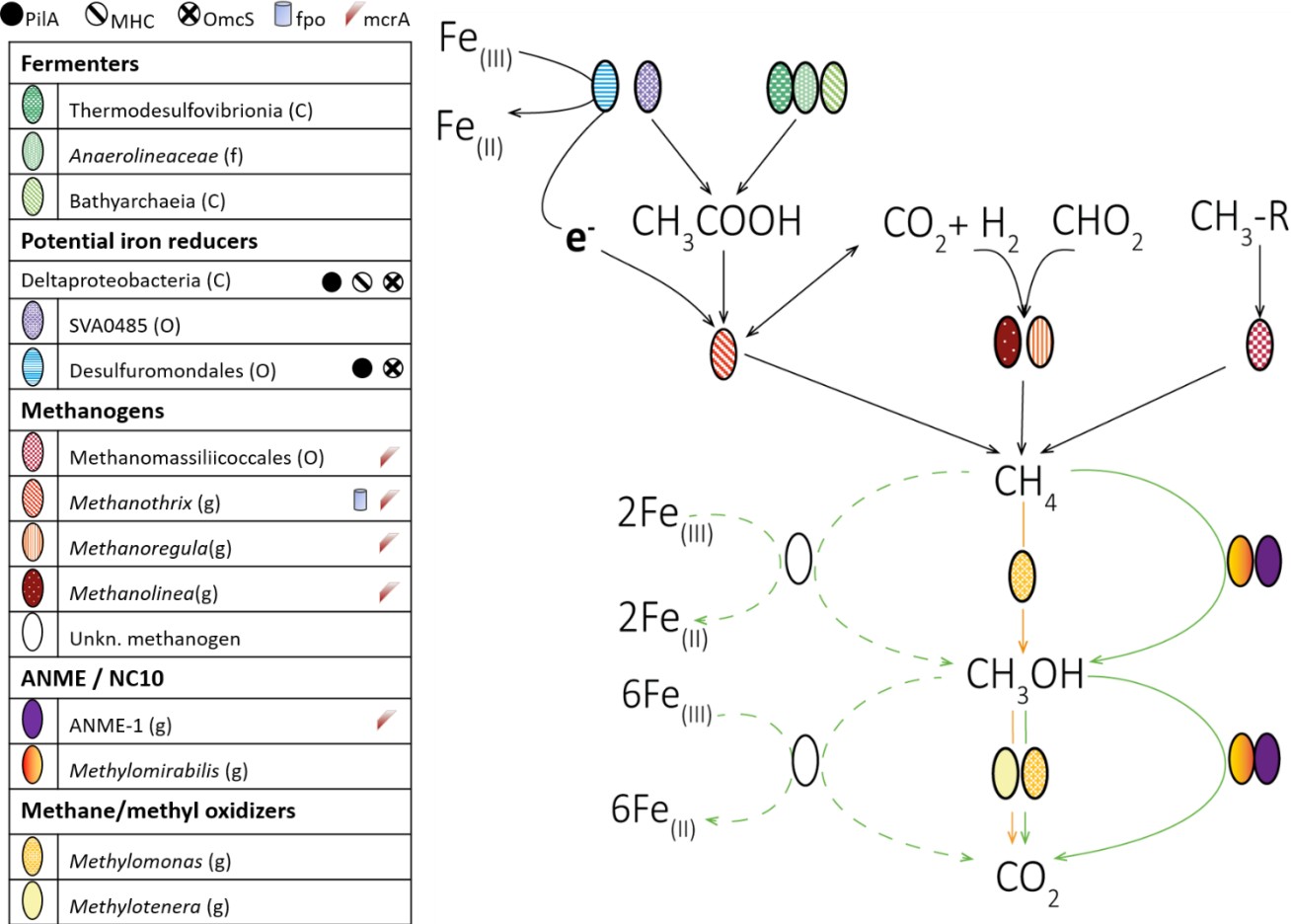

Fig.5. The reaction model of possible methane and iron cycling routes in Lake Kinneret deep sediments (>20 cm sediment depth). Bacteria and Archaea linages are represented by ellipses. Functions associated with iron reduction (MHC=Multi Heme Cytochromes, PilA, OmcS, and the complex Fpo) or methanogenesis (McrA=methyl coenzyme M reductase) are represented by different symbols, and these symbols are shown next to the name of the linages which possess them. Aerobic (orange), anaerobic (green) and unconfirmed anaerobic (dashed green) routes are displayed by arrows. These routes of methane oxidation may occur in parallel. We note that the $CO_2$ can be produced by back flux reactions in methanogenesis. The highest assigned taxonomic levels are shown: Class (C), Order (O), Family (F), Genus (G).

**Data availability:** The metagenome and short reads are available as NCBI BioProject accession number PRJNA637457. The datasets mentioned in the text have been deposited at Figshare open accesses respiratory and can by access at - https://doi.org/10.6084/m9.figshare.c.5245157.v1 (Separate links to each dataset can be found in the supplementary material).

**Competing interests.** The authors declare that they have no conflict of interest.

**Author contribution** OS and ZR designed the project. ME prepared the samples, extracted DNA, analyzed the data, designed and created the figures, and took the lead in writing the manuscript. IBR provided the samples and analyzed the 16S rRNA amplicons. MRB carried the bioinformatics analyses and contributed considerably to the interpretation of the

results and writing of the manuscript. All co-authors provided critical feedback and helped to shape and write the manuscript.

**Acknowledgments** We would like to express our gratitude to all of Orit Sivan's lab members and Zeev Ronen lab technicians - Damiana Diaz and Chen Hargil, for their help in sampling and lab work. A special thanks go to Hanni Vigderovich and Noam Lotem for the helpful scientific discussions and advice. Many Thanks to Ariel Kushmaro, Eitan Ben Dov, Marcus Elvert and Jonathan Groop for their collaboration, assistance, and important insight on the research. We also wish to thank Benny Sulimani and Oz Tzabari from the Yigal Allon Kinneret Limnological Laboratory for their onboard technical assistance. Many thanks to the reviewers for their thorough and constructive reviews. This work was supported by the ERC grant (818450) and the ISF grant (857-2016) to O. S, and the ISF grant (913/19) to M.R-B.

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
