# Peer review of "Metagenomic insights into the metabolism of microbial communities that mediate iron and methane cycling in Lake Kinneret iron-rich methanic sediments"

_Biogeosciences, 2020_

## Referee Comment (RC1) · Anonymous Referee #1 · 23 Sep 2020

The paper addresses a topic of importance to readers of this journal: the microbial ecology of ferruginous sediments. The title is descriptive and therefore does not as clearly summarize the paper's major finding as a declarative title would, but it does accurately describe the paper's topic. The abstract provides a concise and complete summary. The paper is overall well-structured and clearly written, with fluent and precise language, and of appropriate length. The figures are of high quality. The findings largely confirm a previous study (Vuillemin et al 2018), and thus the findings overall are more confirmatory than novel, but important nonetheless.

I have several suggestions for strengthening the methods and results as well as some

missing citations:

1) The paper includes metagenomic data on sediments incubated with various substrates for 470-days but never mentions specifics about the activities of these sediments for methane oxidation, iron reduction, methanogenesis, etc. Please summarize those geochemical data from the Bar-Or et al 2017 study at the start of the results section to set the stage for the metagenomics findings.

2) My second main concern is regarding the methods and results for the PilA proteins, which were identified through a simple KEGG annotation without a detailed analysis necessary to confirm that the aromatic abundance and spacing was sufficient for predicted electroactivity. The authors should add that analysis, as in this paper (https://doi.org/10.1111/1758-2229.12809) to check that the PilA sequences contain the requisite cutoffs for predicted electroactivity ($\geq$9.8% aromatic amino acids, $\leq$22‐aa aromatic gaps and aromatic amino acids at residues 1, 24, 27, 50 and/or 51, and 32 and/or 57) because there are many other type IV-a pilin genes that can easily be mistaken as electroactive PilA. A script is available for calculation of mature pilin length, percent aromatic amino acids and aromatic free gaps (https://github.com/GlassLabGT/Python-script) as described in this paper: https://doi.org/10.1111/1758-2229.12809. Also for the multiheme cytochromes, there are scripts available from a published study: 'cytochrome_stats.py' described in https://doi.org/10.3389/fmicb.2016.00913 and available at https://github.com/bondlab/scripts. Also, note that electroactive PilA are present in lineages outside of Deltaproteobacteria: see https://doi.org/10.1111/1758-2229.12809 https://doi.org/10.1038/ismej.2017.141 and https://doi.org/10.1128/mBio.00579-19.

3) As supplemental data, the authors should include FASTA files with the hits for each of the major genes discussed, so that readers can easily use the sequences, unless the metagenomes have been deposited in annotated form into NCBI. The NCBI BioProject does not contain any genomes with accessions to cite, so it is important for the FASTA

files to be provided with the publication, or else there is no way for readers to locate the new sequences without reprocessing the raw metagenomes in the BioProject PR-JNA637457 (indeed, there are no genomes listed on the BioProject page, so the data are hidden in SRAs, and not easily accessible for BLAST searches). Even better would be to include annotated metagenomes on NCBI and include the assigned NCBI accession numbers in the paper, but currently that is not simple except for metagenome-assembled bins.

4) Consider citing papers by Kelly Wrighton's group on the importance of Candidatus Methanothrix paradoxum for methanogenesis in terrestrial sediments with oxygen exposure. For example: https://doi.org/10.1038/s41467-017-01753-4. Could also help explain the occurrence of genes encoding oxygen-dependent methane monooxygenases if there is occasional oxygen exposure in these sediments. Are they bioturbated?

Specific comments:

Line 40-41: There has been quite a great deal of research on the diversity and metabolic potential of microbial communities in natural anoxic sediments over the past 40 years. I would not characterize this topic as "largely unknown". Please correct language here to focus on a more specific question, perhaps on ferruginous sediments.

L163-164: It is notable that Bathyarchaeia remained one of the dominant lineages even after sediment incubation. It is typical that Bathys quickly "die out" when sealed in bottles for a few weeks-months (for example, https://doi.org/10.1111/gbi.12239) and these were sealed for 470 days! The authors may want to attempt to culture Bathys out of these bottles, since they seem to be persisting, and perhaps even growing.

L205: change "anaerobic conditions" to "anoxic conditions" (metabolisms are anaerobic/aerobic; environments are oxic/anoxic)

L252: correct the misspelling of Methanosarcinales

L287: ORFs per what? Per metagenome?

L301: capitalize "P" in PilA when referring to protein; italicize and lowercase when referring to gene. No such thing as "pilA" non-italicized.

---

## Referee Comment (RC2) · Anonymous Referee #2 · 29 Sep 2020

This study "Metagenomic insights into the metabolism of microbial communities that mediate iron and methane cycling in Lake Kinneret sediments" use metagenomics to investigated microbial communities associated with iron reduction and methane cycling from both natural Lake Kinneret sediments and iron amended slurry incubations. The data and interpretation is generally good. While I find the topic of this study certainly interesting for Biogeosciences, there are several aspects which should be addressed before publication.

Lack of accompanying geochemical analysis, enzyme assay or transcripts analysis make the study descriptive, mostly putative or based on prediction from reference

database in results and discussion. Moreover, metagenomic analysis of fours treatments shows not much different between them or at least the authors didn't present much difference, which question the experiment design or validity of method due to poor coverage of metagenomic method, especially when targeting a minor group in a complex sample.

Metagenomics analysis only covers the ferruginous part of sediment core, so the title, abstract and descriptions throughout the text should be specific, rather than use "whole" lake sediment.

The names of microbes and genes should be in italic, first letter of proteins should be in Capital, please check and correct throughout the whole text.

Line 35 "on average" and "up to" are redundant and not logical here, delete one.

Line 40 "largely unknown" is not precise here, actually there have many studies in recent years, in ferruginous sediments will be more specific

Line 46 change depleting to depleted

Line 71 Diversity of what?

Line 208-211 Did the author measured concentrations of H2 and SO4 in this study? Otherwise, they need to explain how they get these numbers.

---

## Referee Comment (RC3) · Anonymous Referee #3 · 14 Oct 2020

This manuscript on "Metagenomic insights into the metabolism of microbial communities that mediate iron and methane cycling in Lake Kinneret sediments" is very well written and organized. The title accurately describes the subject of the manuscript, though it is a bit dry and lacks any insight into what was concluded in the study. The abstract is clean and concise and effectively summarizes the key findings of the manuscript, which are largely descriptive. The introduction is also well constructed and (mostly) properly referenced, though the statement at line 40 of "largely unknown isn't exactly true. The figures are well put together, informative and high quality with figure 5 a very nice summary of the results/discussion. However, my main concern with this paper is that there is no geochemical data from the incubations to confirm/support the metage-

nomic interpretations. The authors state at line 374 : "our geochemical experiments suggest…." however, no geochemical data is provided. As such, while the authors engage in thorough, well referenced discussion of inferred function based on homology searches, implying that there is experimental geochemical evidence to support their conclusions is misleading unless that data is presented. If it is available it needs to be presented, even if only in the supplement and not the focus of the main text. I find similarity between the in situ sediment samples and all of the incubations for which metagenomes are available to also be curious, especially in the presence of inhibitors. Perhaps some geochemical data could shed some light on this? At line 71-72 the authors state that " slurry incubations……produced substantial amounts of 13C-labelled DIC". How much is "substantial amounts"? Was there iron reduction? H2 production? Or did the slurry just sit there static and are just a reflection of the initial sediment slurry sitting there for over a year, as it sort of looks like from the non-departure from the t0 microbial community (Figure S2). There seems to be some presentation of in situ geochemical data (lines 208-209) though it's unclear if this was measured or a previously reported value. In the absence of any geochemical data, this study is not entirely novel, but rather confirmatory of other studies on the metabolic potential (potential being the key word) of ferruginous sediments (Vuillemin et al. 2018)

---

## Author Comment (AC1) · 31 Oct 2020

Dear anonymous referee #1, We would like to thank you for taking the time to evaluate our paper and for your constructive review and suggestions which helped increase the overall quality of the manuscript. We have carefully considered all your notes and suggestions and revised the manuscript accordingly. We hereby present point-by-point answers to the issues raised (after each comment you will find a response paragraph). We hope that the manuscript will now be suitable for publication in Biogeosciences. Sincerely yours, Michal Elul, on behalf of all co-authors

Anonymous Referee #1: The paper addresses a topic of importance to readers of this

journal: the microbial ecology of ferruginous sediments. The title is descriptive and therefore does not as clearly summarize the paper's major finding as a declarative title would, but it does accurately describe the paper's topic. The abstract provides a concise and complete summary. The paper is overall well-structured and clearly written, with fluent and precise language, and of appropriate length. The figures are of high quality. The findings largely confirm a previous study (Vuillemin et al 2018), and thus the findings overall are more confirmatory than novel, but important nonetheless. I have several suggestions for strengthening the methods and results as well as some missing citations: The paper includes metagenomic data on sediments incubated with various substrates for 470-days but never mentions specifics about the activities of these sediments for methane oxidation, iron reduction, methanogenesis, etc. Please summarize those geochemical data from the Bar-Or et al 2017 study at the start of the results section to set the stage for the metagenomics findings.

response: We thank the referee for this observation and agree that a summary presenting the geochemical data on Bar-Or et al 2017 slurries is needed. In the revised version, as recommended, we devoted a section (3.1) at the beginning of the results for this purpose. In this section, we describe the concentrations of relevant elements (methane, dissolved iron, manganese, nitrate, and sulfate) in the investigated sedimentary zone as well as the geochemical data on methane oxidation, iron reduction, and methanogenesis processes in Bar-Or et al 2017 slurries. We have attached to this response form the above mentioned new section (3.1).

2) My second main concern is regarding the methods and results for the PilA proteins, which were identified through a simple KEGG annotation without a detailed analysis necessary to confirm that the aromatic abundance and spacing was sufficient for predicted electroactivity. The authors should add that analysis, as in this paper (https://doi.org/10.1111/1758-2229.12809) to check that the PilA sequences contain the requisite cutoffs for predicted electroactivity ($\geq$9.8% aromatic amino acids, $\leq$22âAËŸ Raa aromatic gaps, and aromatic amino acids
at ËĞ residues 1, 24, 27, 50 and/or 51, and 32 and/or 57) because there are many other type IV-a pilin genes that can easily be mistaken as electroactive PilA. A script is available for calculation of mature pilin length, percent aromatic amino acids and aromatic free gaps (https://github.com/GlassLabGT/Pythonscript) as described in this paper: https://doi.org/10.1111/1758-2229.12809. Also for the multiheme cytochromes, there are scripts available from a published study: 'cytochrome_stats.py' described in https://doi.org/10.3389/fmicb.2016.00913 and available at https://github.com/bondlab/scripts. Also, note that elec-troactive PilA are present in lineages outside of Deltaproteobacteria: see https://doi.org/10.1111/1758-2229.12809 https://doi.org/10.1038/ismej.2017.141 and https://doi.org/10.1128/mBio.00579-19

response: As suggested, we confirmed that the aromatic abundance and spacing was sufficient for predicted electroactivity in the metagenome pilA sequences using the recommended script. We corrected Figure 4d, which now shows only the PilA open reading frames that correspond to the stringent parameters (the amended figure 4d in attached to the response form). Accordingly, we adjusted the text in this paragraph to: " The overall abundance of the MHC (secreted and trans-membranal), PilA and OmcS ORFs was 364-493, 35-45, 5-9 and 4-9 counts per million reads mapped, re-spectively. Our findings confirm that the phylogenetic diversity of microbes are capable of nanowire-mediated DIET extends beyond deltaproteobacterial lineages (Bray et al. 2020), as strict searches attributed pilA-like sequences not only to Desulfobacterota (Deltaproteobacteria), but also to Thermodesulfovibrionales, Burkholderiales, Gemma-timonadales, Aminicenantales, as well as WOR-3 and Firmicutes (Fig, 4d)." We thank the reviewer for pointing out the 'cytochrome_stats.py' script, this will streamline our future analyses.

3)As supplemental data, the authors should include FASTA files with the hits for each of the major genes discussed, so that readers can easily use the sequences, unless the metagenomes have been deposited in annotated form into NCBI. The NCBI Bio-

Project does not contain any genomes with accessions to cite, so it is important for the FASTA files to be provided with the publication, or else there is no way for readers to locate the new sequences without reprocessing the raw metagenomes in the BioProject PRJNA637457 (indeed, there are no genomes listed on the BioProject page, so the data are hidden in SRAs, and not easily accessible for BLAST searches). Even better would be to include annotated metagenomes on NCBI and include the assigned NCBI accession numbers in the paper, but currently that is not simple except for metagenome assembled bins.

Response: As suggested by the reviewer, we submitted the metagenome to NCBI within the PRJNA637457 project. The metagenome is currently being processed and will be released ASAP. We supplemented the manuscript with amino acid sequences of the enzymes discussed (those involved in methanogenesis and extracellular electron transfer, as well as heterodisulfide reductase ‎subunits) in FASTA files, referred to as Supplementary Database 6, 7, and 8. The above-mentioned FASTA files can be found in the following links: (which are also listed as S.DB.6,7 and 8 in the revised version of the Supplementary Information).

S.DB.6| Metagenomic hits (amino acid sequences) for methanogenesis related enzymes -FwdC/FmdC, Ftr, Mch , MtrA, Mer, Mtd, mcrA. https://doi.org/10.6084/m9.figshare.13091126.v2

S.DB.7| Metagenomic hits (amino acid sequences) for extracellular electron transfer‎ related enzymes - MHC, omcS and pilA. https://doi.org/10.6084/m9.figshare.13092821.v1

S.DB.8| Metagenomic hits (amino acid sequences) for heterodisulfide reductase ‎subunits A, D and E. https://doi.org/10.6084/m9.figshare.13092842.v1

4) Consider citing papers by Kelly Wrighton's group on the importance of Candidatus Methanothrix paradoxum for methanogenesis in terrestrial sediments with oxygen exposure. For example: https://doi.org/10.1038/s41467-017-01753-4. Could also help

explain the occurrence of genes encoding oxygen-dependent methane monooxyge-nases if there is occasional oxygen exposure in these sediments. Are they bioturbated?

Response: We thank the reviewer for this information. We assume that Lake Kineret sediments are not bioturbated in the depths that we examined (26-41cm). We now cite a paper from Kelly Wrighton's group (https://doi.org/10.1038/s41467-017-01753-4): "Other notable archaeal lineages included the acetoclastic Methanothrix (1-3% read abundance), which are often found en masse in anoxic lake sediments (Smith and Ingram-Smith 2007; Schwarz et al. 2007; Carr et al. 2018) as well as in oxygenated soil, as recently discovered for Methanothrix paradoxum (Angle et al. 2017)."

Specific comments: Line 40-41: There has been quite a great deal of research on the diversity and metabolic potential of microbial communities in natural anoxic sediments over the past 40 years. I would not characterize this topic as "largely unknown". Please correct language here to focus on a more specific question, perhaps on ferruginous sediments.

Response: We agree with the reviewer that this line needed to be more focused on a specific topic. The text now reads "However, the diversity and metabolic potential of the microbial communities in natural anoxic ferruginous sediments are not fully understood"

L163-164: It is notable that Bathyarchaeia remained one of the dominant lineages even after sediment incubation. It is typical that Bathys quickly "die out" when sealed in bottles for a few weeks-months (for example, https://doi.org/10.1111/gbi.12239) and these were sealed for 470 days! The authors may want to attempt to culture Bathys out of these bottles, since they seem to be persisting, and perhaps even growing.

Response: We are thrilled to try it!

L205: change "anaerobic conditions" to "anoxic conditions" (metabolisms are anaerobic/aerobic; environments are oxic/anoxic).

Response: Corrected as suggested.

L252: correct the misspelling of Methanosarcinales.

Response:Corrected as suggested.

L287: ORFs per what? Per metagenome?

Response: Indeed, per metagenome, we added this clarification in the text.

Please also note the supplement to this comment:
https://bg.copernicus.org/preprints/bg-2020-329/bg-2020-329-AC1-supplement.pdf

———————————————————

[Figure]

(a)

MHC transmembrane

| | |
|---|---|
| t0 | 35 |
| +$^{13}CH_4$ | 45 |
| $^{13}CH_4$+Hematite | 39 |
| $^{13}CH_4$+A.Fe(III)+$MoO_4$ | 43 |

(b)

MHC secreted

| | |
|---|---|
| t0 | 364 |
| +$^{13}CH_4$ | 493 |
| $^{13}CH_4$+Hematite | 396 |
| $^{13}CH_4$+A.Fe(III)+$MoO_4$ | 439 |

(c)

OmcS

| | |
|---|---|
| t0 | 4 |
| +$^{13}CH_4$ | 9 |
| $^{13}CH_4$+Hematite | 7 |
| $^{13}CH_4$+A.Fe(III)+$MoO_4$ | 9 |

(d)

PilA

| | |
|---|---|
| t0 | 4 |
| +$^{13}CH_4$ | 9 |
| $^{13}CH_4$+Hematite | 7 |
| $^{13}CH_4$+A.Fe(III)+$MoO_4$ | 5 |

Read abundance [%]

Legend:
- ■ Proteobacteria;Deltaproteobacteria;Desulfuromonadales
- ■ Proteobacteria;Deltaproteobacteria;Myxococcales
- ■ Proteobacteria;Deltaproteobacteria
- ■ Proteobacteria;Betaproteobacteria;Burkholderiales
- ■ Proteobacteria
- ■ Acidobacteria;Aminicenantia;Aminicenantales
- ■ Acidobacteria
- ■ Gemmatimonadetes;Gemmatimonadetes;Gemmatimonadales
- ■ Actinobacteria
- ■ Chloroflexi
- ■ Firmicutes
- ■ Nitrospirae;Thermodesulfovibrionia;Thermodesulfovibrionales
- ■ Nitrospirae
- ■ WOR-3
- ■ Unclassified Bacteria
- ■ Unclassified
- ■ Order (or higher rank) ≤3%

**Fig. 1.**

---

## Author Comment (AC2) · 31 Oct 2020

Dear anonymous referee #2 We thank you for the thorough and thoughtful comments on our submitted article. We went through the comments and suggestions and the paper has been revised accordingly. We present below, point-by-point, answers to the issues raised (after each comment you will find a response paragraph). We hope that you will find the revised version of our manuscript suitable for publication in Biogeosciences.

Sincerely yours, Michal Elul, on behalf of all co-authors

[Figure]

Anonymous Referee #2: This study "Metagenomic insights into the metabolism of microbial communities that mediate iron and methane cycling in Lake Kinneret sediments" use metagenomics to investigated microbial communities associated with iron reduction and methane cycling from both natural Lake Kinneret sediments and iron amended slurry incubations. The data and interpretation is generally good. While I find the topic of this study certainly interesting for Biogeosciences, there are several aspects which should be addressed before publication. Lack of accompanying geochemical analysis, enzyme assay or transcripts analysis make the study descriptive, mostly putative or based on prediction from reference database in results and discussion.

Response: We agree with the reviewer that a geochemical background and analyses of both the sedimentary zone and slurry incubations examined here is needed to be added (also noticed by reviewers 1 and 3). In the revised version, we added a full section (3.1) that address the geochemical aspect of the manuscript. This new section is attached to this response form. Since our study was based on metagenomics, it can only raise hypotheses regarding the functionality of the studied communities. We agree that further experiments, such as enzyme assay or metatranscriptomics are needed to base our assumptions. We strongly believe, however, that this study provides a valuable basis for further investigation of Lake Kinneret communities and iron and methane metabolisms.

Moreover, metagenomic analysis of four treatments shows not much different between them or at least the authors didn't present much difference, which question the experiment design or validity of method due to poor coverage of metagenomic method, especially when targeting a minor group in a complex sample.

Response: Albeit the overall similarities, we find some differences between the treatment, yet lack the statistical power to show them and often can only speculate regarding their nature. For example, BES additions appear to reduce the relative abundance of Methanosarcinales, but not Methanomicrobiales, as observed in the 16S rRNA amplicon read results. We agree that the small changes following the addition of BES are

curious, yet at this point, we prefer not to overinterpret these changes. Iron mineral amendments may have little effect on the community structure, as iron are not limiting in these sediments. The overall similarity of the communities allowed us to increase the coverage and co-assemble the reads from the different libraries, being in our favor in this case. We believe that although the coverage was insufficient to cover the rare taxa in the way that high-quality bins could be assembled, metagenome-wide functional predictions and taxonomic assignments still provided important insights into this system.

Metagenomics analysis only covers the ferruginous part of sediment core, so the title, abstract and descriptions throughout the text should be specific, rather than use "whole" lake sediment.

Response: We agree - in the revised version we emphasize in both the title, abstract and descriptions throughout the text that our analyses address only the deep iron-rich methanic part of the sediment in Lake Kinneret. The title has been changed to "Metagenomic insights into the metabolism of microbial communities that mediate iron and methane cycling in Lake Kinneret iron-rich methanic sediments".

The names of microbes and genes should be in italic, first letter of proteins should be in Capital, please check and correct throughout the whole text

Response: Thank you for these observations, we made amendments throughout the text accordingly.

Specific comments: Line 35 "on average" and "up to" are redundant and not logical here, delete one.

Response: Corrected as suggested.

Line 40 "largely unknown" is not precise here, actually there have many studies in recent years, in ferruginous sediments will be more specific.

Response: This sentence now reads: "However, the diversity and metabolic potential of the microbial communities in natural anoxic ferruginous sediments are not fully understood."

Line 46 change depleting to depleted

Response: Corrected as suggested.

Line 71 Diversity of what?

Response: We refer to the diversity of bacteria and archaea. For clarity, this line now reads: "In all the treatments, the diversity of bacteria and archaea was similar to that of the natural sediments"

Line 208-211 Did the author measured concentrations of H2 and SO4 in this study? Otherwise, they need to explain how they get these numbers.

Response: Our group measured these species. H2 concentrations were measured by Michal Adler and shown in her doctoral dissertation, and SO4 concentrations were measured in Adler et al. 2011;Sivan et al.2011 and Bar-Or et al., 2015. The references were added as suggested.

Please also note the supplement to this comment:
https://bg.copernicus.org/preprints/bg-2020-329/bg-2020-329-AC2-supplement.pdf

**3.1 Geochemical evidence for iron coupled AOM in Lake Kinneret iron-rich methanic sediments**

We explore here slurries amended with Lake Kinneret sediments from the deep methanic zone (26-41 cm). In this potentially ferruginous zone, sedimentary profiles show that the concentration of methane decreases from its maximum values of above 2mM at around 10 cm depth to 500 μM at 40 cm depth, and that of dissolved ferrous iron increases (from 1-6μM at the first 10 cm depth to ~60-100μM, depending on sampling season). This, combined with an increase of $\delta^{13}C$ of methane (from -65‰ at 7 cm depth to -53.5‰ at 24 cm depth) and a decrease of $\delta^{13}C$ of total lipid compounds (from 27‰ at 23 cm depth to -31‰ at 27 cm depth), suggests AOM in the deep sediment coupled to iron reduction (Adler et al. 2011; Sivan et al. 2011). This was supported by rate modeling and by microbial profiles (Adler et al. 2011; Sivan et al. 2011; Bar-Or et al. 2015, 2017). Alternative electron acceptors are scarce: dissolved manganese oxides concentrations are ~ 0.04% and nitrate and sulfate are below the detection limit (Sivan et al. 2011).

The slurries investigated microbially here were amended with isotopically labeled $^{13}CH_4$, $^{13}CH_4$ + hematite and $^{13}CH_4$ + amorphous iron + molybdate for 470 days. In these incubations, we observed a marked enrichment of labeled carbon after ten months of incubation (up to 250‰ enrichment in the treatment with hematite addition, up to 80‰ enrichment in the natural treatment and up to 450‰ in the treatment with amorphous iron + molybdate Fig. S1 in the Supplement). Ferrous iron concentrations increased by ~20−50 μM following iron oxide amendments (with and without molybdate addition), indicating that iron was reduced. The BES amendments resulted in the highest increase in ferrous iron concentrations (~50-110 μM), most likely due to the abiotic reaction of BES with iron minerals. The evidence for iron reduction, together with the fact that $\delta^{13}C_{DIC}$ values increased by 250-450‰ in the different iron amended treatments, but not in methane-only additions (only up to 80‰, Fig. S1 in the Supplement), indicate iron coupled AOM. Sulfate did not play a role in the AOM, as the addition of molybdate, sulfate reduction and disproportionation antagonist, did not inhibit methane turnover (Fig. S1 in the Supplement). The addition BES to specific slurries inhibited the production of $\delta^{13}C_{DIC}$, indicating the essential role of methanogens in the AOM activity (Fig. S1 in the Supplement).

**Fig. 1.**

---

## Author Comment (AC4) · 31 Oct 2020

Dear anonymous referee #3 Thank you for taking the time to assess our manuscript. We appreciate your valuable comments, suggestion and corrections. We have carefully addressed each concern raised and revised the manuscript accordingly. We hereby present point-by-point answers to the issues raised (after each comment you will find a response paragraph). We hope that the manuscript will now be suitable for publication in Biogeosciences. Sincerely yours, Michal Elul, on behalf of all co-authors.

Anonymous Referee #3 This manuscript on "Metagenomic insights into the metabolism of microbial communities that mediate iron and methane cycling in Lake Kinneret sediments" is very well written and organized. The title accurately describes the subject of the manuscript, though it is a bit dry and lacks any insight into what was concluded in the study. The abstract is clean and concise and effectively summarizes the key findings of the manuscript, which are largely descriptive.

Response: We thank the reviewer for the positive feedback.

The introduction is also well constructed and (mostly) properly referenced, though the statement at line 40 of "largely unknown isn't exactly true.

Response: Following the recommendation of all three reviewers this line was changed to "However, the diversity and metabolic potential of the microbial communities in natural anoxic ferruginous sediments are not fully understood"

However, my main concern with this paper is that there is no geochemical data from the incubations to confirm/support the metagenomic interpretations. The authors state at line 374 : "our geochemical experiments suggest: : :." however, no geochemical data is provided. As such, while the authors engage in thorough, well referenced discussion of inferred function based on homology searches, implying that there is experimental geochemical evidence to support their conclusions is misleading unless that data is presented. If it is available it needs to be presented, even if only in the supplement and not the focus of the main text.

Response: We thank the reviewer for this helpful comment. In the revised manuscript, we added a new section (3.1) that briefly addresses the geochemistry of the sampled sediments and the geochemical analyses of the slurry incubations. We supplement this discussion with figure S1 in the Supplement, which shows the change in $\delta 13C$ of the DIC of the slurry incubations, after Bar-Or et al.2017. Both the new section (3.1) and figure S1 are attach to this response form.

I find similarity between the in situ sediment samples and all of the incubations for which metagenomes are available to also be curious, especially in the presence of

inhibitors. Perhaps some geochemical data could shed some light on this?

Response: We have observed some dissimilarities between the treatments, however, our analyses lack the statistical power to clearly define these differences. We can speculate that iron amendments had little effect on the composition of microbial communities, as iron is not a limiting factor in these sediments. Similarly, as we suspect that sulfate plays only a minor role in these sediments due to the low concentrations, the addition of molybdate may have only a negligible effect on the community structure. Bar Or et al. 2017 geochemical data (now presented as Supplementary Figure S1) shows that the addition of BES completely halted methanotrophy and methanogenesis. We observed that read abundance of some lineages, such as Methanosarcinales, declined in BES amendments (Supplementary Figure S2, S1 in the previous version). It is still unclear how other methanogens persist in BES-amended treatments, transcriptomics may elucidate this interesting phenomenon. It is important to note that the results here describe only the relative abundance. It is feasible that the cell numbers declined following the BES addition. In this study, the fact that the communities are similar among the treatments is, in fact, helpful for our analyses, allowing co-assembly and thus better genomic coverage.

At line 71-72 the authors state that " slurry incubations: : :: : :produced substantial amounts of 13C-labelled DIC". How much is "substantial amounts"?

Response: We clarify this in the text and refer to the new Supplementary Figure 1: "These incubations, including a) 13CH4, b) 13CH4 + Hematite, or c) 13CH4 + amorphous iron + molybdate (A.Fe(III)+MoO4) produced substantial amounts of 13C-labelled dissolved inorganic carbon over 470 days (80-450‰ Fig. S1 in the Supplement)". As stated above, we added section 3.1 to introduce the geochemical data.

Was there iron reduction? H2 production? Or did the slurry just sit there static and are just a reflection of the initial sediment slurry sitting there for over a year, as it sort of looks like from the non-departure from the t0 microbial community (Figure S2).

Response: Iron reduction occurred in the slurry incubations. We address this sub-ject in the newly added section 3.1-L154 " Ferrous iron concentrations increased by âĹij20−50 $\mu$M following iron oxide amendments (with and without molybdate addition), indicating that iron was reduced." Unfortunately, H2 was not measured in the slurry incubations.

There seems to be some presentation of in situ geochemical data (lines 208-209) though it's unclear if this was measured or a previously reported value.

Response: The values mentioned here are previously reported values. To clarify this issue, the respective references we added: The hydrogen concentration in the Fe-AOM horizon is ∼20 $\mu$M gr-1 sediment (Adler 2015). Given that sulfate is below the detection limit there (<10$\mu$M, Adler et al., 2011, Sivan et al., 2011), hydrogen scavenging may also be coupled to metal reduction, most likely by Deltaproteobacterial lineages, some of which may be syntrophic (e.g. Syntrophobacterales). "
* * *
**3.1 Geochemical evidence for iron coupled AOM in Lake Kinneret iron-rich methanic sediments**

We explore here slurries amended with Lake Kinneret sediments from the deep methanic zone (26-41 cm). In this potentially ferruginous zone, sedimentary profiles show that the concentration of methane decreases from its maximum values of above 2mM at around 10 cm depth to 500 µM at 40 cm depth, and that of dissolved ferrous iron increases (from 1-6µM at the first 10 cm depth to ~60-100µM, depending on sampling season). This, combined with an increase of $\delta^{13}C$ of methane (from -65‰ at 7 cm depth to -53.5‰ at 24 cm depth) and a decrease of $\delta^{13}C$ of total lipid compounds (from 27‰ at 23 cm depth to -31‰ at 27 cm depth), suggests AOM in the deep sediment coupled to iron reduction (Adler et al. 2011; Sivan et al. 2011). This was supported by rate modeling and by microbial profiles (Adler et al. 2011; Sivan et al. 2011; Bar-Or et al. 2015, 2017). Alternative electron acceptors are scarce: dissolved manganese oxides concentrations are ~ 0.04% and nitrate and sulfate are below the detection limit (Sivan et al. 2011).

The slurries investigated microbially here were amended with isotopically labeled $^{13}CH_4$, $^{13}CH_4$ + hematite and $^{13}CH_4$ + amorphous iron + molybdate for 470 days. In these incubations, we observed a marked enrichment of labeled carbon after ten months of incubation (up to 250‰ enrichment in the treatment with hematite addition, up to 80‰ enrichment in the natural treatment and up to 450‰ in the treatment with amorphous iron + molybdate Fig. S1 in the Supplement). Ferrous iron concentrations increased by ~20−50 µM following iron oxide amendments (with and without molybdate addition), indicating that iron was reduced. The BES amendments resulted in the highest increase in ferrous iron concentrations (~50-110 µM), most likely due to the abiotic reaction of BES with iron minerals. The evidence for iron reduction, together with the fact that $\delta^{13}C_{DIC}$ values increased by 250-450‰ in the different iron amended treatments, but not in methane-only additions (only up to 80‰, Fig. S1 in the Supplement), indicate iron coupled AOM. Sulfate did not play a role in the AOM, as the addition of molybdate, sulfate reduction and disproportionation antagonist, did not inhibit methane turnover (Fig. S1 in the Supplement). The addition BES to specific slurries inhibited the production of $\delta^{13}C_{DIC}$, indicating the essential role of methanogens in the AOM activity (Fig. S1 in the Supplement).

**Fig. 1.**

[Figure]

**Figure S1:** Net change in $\delta^{13}C_{DIC}$ values in Bar-or et al.2017 slurry incubations after 470 days. Solid blue: without inhibitor addition; Fence red: inhibition of methanogenesis by BES addition; Dot green: inhibition of sulfate reduction by molybdate addition. Analysis of DNA 16S rRNA genes was performed for all of these incubations and the untreated sediments. The following treatments: Natural (without additions), Amorphous iron with the addition of molybdate and the hematite (without additions) treatment were sequenced for metagenome analysis.   After Bar-Or et al. 2017

**Fig. 2.**

---

## Referee Comment (RC4) · Anonymous Referee #4 · 6 Nov 2020

Summary:

The manuscript by Elul et al reports the results of 16s amplicon and shotgun metagenomic analysis of a narrow sediment horizon from Lake Kinneret. These DNA analyses were conducted on freshly sampled sediment and sediment that had undergone the incubations characterized in detail in Bar-Or et al 2017. The authors focus their attention on enzyme systems that may be associated with iron or methane cycling. The authors provide information on the phylogenetic composition of the microbial community in general, as well as assign phylogenetic composition to specific enzymes by BLASTing the metagenome reads against the refseq database.

[Figure]

Major concerns:

1) Insufficient information is given about the incubations which is needed to fully evaluate the likelihood of the conclusions presented in the current work (most crucially, these incubations are methanogenic).

2) The suggestion that Methanothrix may carry out a methane oxidizing metabolism breaks with everything that is known about this group, and the claim is not supported by any experimental data. This suggestion should be removed.

3) The authors do not carry out any calculations to support their claim that traditional ANME are not abundant enough to carry out the trace AOM they claim to observe, and no effort is made to engage with the thermodynamic feasibility of the processes they are proposing, which is fairly straightforward and should be done.

Concerns 1&2:

This manuscript is framed as a study that will draw significant insight from incubations. Incubations with specific substrates or inhibitors can be very powerful tools in environmental microbiology, particularly when the microbial community responds to the incubation conditions, and when care is taken to clearly describe the bulk geochemical processes that have occurred in the incubations. Unfortunately, this is not the case in this study, while I understand that the bulk of the description of the incubations was previously published, a few key pieces of information have been left out of the current manuscript.

It would likely appear to a reader that these are incubations in which methane oxidation is the dominant process since so much emphasis is put on AOM as compared to methanogenesis. AOM is the most discussed metabolism in the abstract, and a major conclusion is the surprising attribution of AOM metabolism to Methanothrix. However, these incubations are NOT carrying out the net oxidation of methane, they are net methanogenic (see Figure 2b of Bar-Or 2017 "Positive methane concentrations reflect

net methanogenesis during iron-coupled AOM.").

To put the results more plainly: sequencing of methanogenic incubations reveals a dominant archaeon that is a well-known methanogen. When stated in this way, I cannot support the publication of such a speculative assignment of AOM activity to Methanothrix. The simplest explanation is that the dominant methanogen is growing via the dominant methane cycling process, i.e. methanogenesis.

The justification for any discussion of AOM relies heavily on the previous publication that found 13C methane was converted into 13C CO2, and this activity was inhibited by BES. Methanogens carry out backflux of isotopic label from methane to CO2, and the authors have cited the classic paper that shows this (Zehnder and Brock, 1979). Methanothrix could indeed be responsible for the conversion of 13C methane into 13C CO2, but this observation does not constitute evidence that they carry out net AOM in the environment or in these incubations. It is crucially important for metabolisms that are so close to equilibrium for the authors to be very clear about whether they are suggesting an organisms is making energy for growth by carrying out AOM, or whether the organism may simply be responsible for the equilibration of isotope labels in the opposite direction of the process they are using for energy generation.

Another line of evidence for AOM is reaction-diffusion modeling that was carried out on Lake Kinneret sediments (Adler et al 2011), which concluded that there was peak methanogenesis 5-12cm below the sediment surface, and there was deeper AOM region under that. But microbial 16s profiling carried out in Bar-Or et al 2015, did not show a significant change of methanothrix (there referred to as methanosaeta) between the methanogenic and the methane oxidation zones.

This is a big claim the authors are trying to make, and it would require some sort of direct evidence like: 1) if there was an incubation where AOM was the dominant processes and the authors were able to show that methanothrix was the only organism present with the seven step methanogenesis pathway; 2) or better yet that methanothrix was enriched under these conditions vs. conditions without methane/Fe addition; 3) or, upon the addition of methane (and Fe?) there was a positive reaction of methanothrix based on metatranscriptome analyses, 4) or, at the very least that in nature there was a correlation between methanothrix abundance and the horizons where methane oxidation is occurring.

Unfortunately, the community did not significantly change under any incubation condition (line 45), and there is no correlation presented from the natural distribution of species, so there is no valid justification for assigning a novel role to an organism that could just be making methane. Unless stronger evidence exists, all claims like the one in line 375: "Our data hints that Methanothrix, which has not been considered to be involved in Fe-AOM previously, has the potential to be involved in methane oxidation, as presented in figure 5" should be removed.

Concern 3:

If the authors reject the isotope backflux idea (there is not a clear quantitative argument against this, even in Bar-Or et al 2017), and insist that there must be an organism subsisting on AOM in their incubations, then it is unclear why the minor, traditional ANME organisms will not suffice.

In the abstract the authors write (lines 23-24) that "bonafide [sic] anaerobic oxidizers of methane (ANME) and denitrifying methanotrophs Methylomirabilia (NC10) were scarce", discounting their role in AOM in these sediments. But then they highlight on line 25-26 "We show that putative aerobes, such as methane-oxidizing bacteria Methylomonas and their methylotrophic syntrophs methylotenera... can be involved in the oxidation of methane...".

It is not at all clear why the authors feel that ANME should be discounted while aerobic methanotrophs should be accepted as being responsible for methane oxidation. On line 176 the authors say that 0.3-0.8% of their reads map to ANME-1. And the very next paragraph the authors discuss the type I methanotrophs which are found to be 0.4-

1.8% of the community. There is no meaningful difference between 0.3-0.8% and 0.4-1.8% in terms of abundance, so why do they feel comfortable highlighting the possible role of aerobic methanotrophs at this abundance and not the anaerobic ones? Why have the aerobic methane oxidizers made it into Fig 5 but the bona fide ANME have not?

AOM is not the dominant process, so its seems reasonable that if there is a small methane oxidizing community that it could be carried out by normal methane oxidizers that are in low abundance. The only way to rule this out is to determine the rate of AOM, try to estimate what 0.3-0.8% read mapping may correspond to in terms of cell numbers, and then calculate a cell specific rate and show that this rate seems far too high when compared to values present in the literature for ANME rates. None of this work is done.

When discussing possible metabolisms and their putative relative importance, it is very helpful to discuss the thermodynamic feasibility of these reactions. But in the summary line 380-381 the authors write "...whether this process [methanothrix AOM] is justified from the thermodynamic and kinetic perspectives, remains to be elucidated.". Doing the thermodynamic analysis should be a bare minimum requirement when suggesting a remarkable new metabolism for an organism. What are the relative free energies associated with acetoclastic methanogenesis and then Fe-AOM vs. acetate oxidizing iron reduction? For a study that is essentially just a single metagenomic analysis (since there is no noteworthy difference between any of the samples), the authors should at least attempt to supplement their discussion with thermodynamic discussions.

Minor comments:

"Consortium" should not be used interchangeably with "community" especially in the context of AOM research where "consortium" is very commonly used to refers to a physical, presumably syntrophic association between two microorganisms. Since no evidence is provided about actually association between any organisms described in

this study "consortium" should be replaced throughout with "community".

Line 361: "Our results show that in general, the phylogenetic diversity is a good predictor of the functional diversity in these samples". This is too broad of a statement for a paper that has a fairly narrow focus on iron and methane cycling.

Line 20: not clear what "intrinsic" means in this context. Are any organisms in this sample not intrinsic?

Line 63: Assigning Thermodesulfovibrio to a carbon oxidizing, iron reducing metabolism is wildly speculative and should be removed unless more work is done to support the claim. The authors cite Spring et al 1993 (indirectly, by way of Bar-Or et al 2015) for this claim. Spring et al does not make this claim, they suggest as a throw-away hypothetical in the discussion section that it could be possible that Magnetobacterium could gain energy from sulfide oxidation coupled to iron reduction. They had no evidence for that claim, just suggested it was possible because Magnetobacterium has magnetosomes and lives in sulfidic environments. If the authors want to follow up this speculation with analysis, then they could look for the magnetosome genes in their metagenomes and see if they are phylogenetically aligned with Magnetobacterium (see Lin et al 2014 for the genes in magnetobacterium, https://www.nature.com/articles/ismej201494). If these thermodesulfovibrio have magnetosomes then maybe its worth mentioning this, but even then, it is probably worth noting that there is no actual evidence that these organisms can grow in this way.

Line 143-145: Here the use of "limiting nutrient" is confusing. This term often refers to something that is a growth requirement because it is needed for the production of biomolecules or cofactors, P, N, Fe, etc. This is a different concept than iron being used for the purpose of an electron acceptor, which seems to be the focus of this study. Clarification is needed.

Line 151: three groups are listed and then "3-6% read abundance, respectively". Incorrect usage of respectively, not clear what each groups abundance is.

[Figure]

Line 158: class-level phylogenetic information should not be taken as evidence for the pH optimal for a group (the authors actually site a paper that describes how a different species of thermodesulfovibrio is alkaliphilic as compared to other species in that genera). This is definitely is not evidence for acidic/basic microenvironments.

Line 378: "positive correlation between Methanosarcinales abundance and concentrations of reduced iron in the deep sediment sections (Bar-Or et al 2017)". This is a very strange claim and I cannot find any significant data that supports it. Bar-Or 2017 does not include pore water profiles or depth profiles of Methanosarcinales, so maybe this reference is supposed to be Bar-Or et al 2015? Even so, the data presented in Bar-Or et al 2015 Figure 4 is single replicate from three depth points. It looks like the difference between 6-9cm and 29-32cm for methanosarcinales is 50% -> 55% at most? With this level of replication this is not a significant correlation that should be taken as evidence supporting methanosarcinales being responsible for iron reduction.

Figure 4: something is wrong with the description, or the data presented. For OmcS LK-2017 the number next to the bar is 4, which the caption says corresponds to the number of total reads mapped to a gene. That bar shows very fine delineations, "Deltaproteobacteria" is maybe 1/20th of the total area of the bar? How can you get 1/20th with only 4 reads mapped? This comment applies to other bars in the OmcS figure. Maybe worth revisiting how these were calculated?

Line 389: "Another possible explanation for the methylated compound leakage is the reversibility of the enzymes involved in AOM, in particular methyl-CoM reductase". Mcr does not may methylated compounds like the ones the authors are referring to in the forward or reverse direction, so the reversibility of this enzyme has nothing to do with this discussion.

Figure 5. The schematic in the top left shows iron reduction (Fe(III) -> Fe(II)) producing electrons

---

## Author Comment (AC5) · 26 Nov 2020

Dear anonymous referee #ŐŔ4Ŕ We appreciate the time and effort that you dedicated to providing feedback on our manuscript and are Őgrateful for the insightful comments and suggestions. We hereby present point-by-point answers to the Őissues raised (after each comment you will find a response paragraph).ŐŔ Ŕ We hope that the manuscript will now be suitable for publication in BiogeosciencesŔ.Ŕ Sincerely yoursŔ,Ŕ Michal Elul, on behalf of all co-authorsŔ Ŕ

Rev#4Ő The manuscript by Elul et al reports the results of 16s amplicon and shotgun metagenomic analysis of a Őnarrow sediment horizon from Lake Kinneret. These DNA analyses were conducted on freshly sampled Ősediment and sediment that had undergone the incubations characterized in detail in Bar-Or et al 2017. The Őauthors focus their attention on enzyme systems that may be associated with iron or methane cycling. The Őauthors provide information on the phylogenetic composition of the microbial community in general, as Őwell as assign phylogenetic composition to specific enzymes by BLASTing the metagenome reads against Őthe RefSeq database.Ő

Response: We thank the reviewer for this thorough review. Ő

Major concerns:  $\hat{a}A\tilde{O} \hat{a}A\tilde{O}1$ ) Insufficient information is given about the incubations which is needed to fully evaluate the likelihood of  $\hat{a}A\tilde{O}$ the conclusions presented in the current work (most crucially, these incubations are methanogenic).  $\hat{a}A\tilde{O}$  Response: As requested by all the referees, we added section 3.1, named "Geochemical  $\hat{a}A\tilde{O}$ evidence for iron coupled AOM in Lake Kinneret iron-rich methanic sediments". In this  $\hat{a}A\tilde{O}$ section, we describe the change in ferrous iron,  $\delta 13$ CDIC and methane concentrations with  $\hat{a}A\tilde{O}$ time in the incubations. This section also includes a description of the concentrations of  $\hat{a}A\tilde{O}$ investigated sedimentary methanogenic zone. We added also that these incubations are  $\hat{a}A\tilde{O}$ indeed methanogenic (see more below).  $\hat{a}A\tilde{O}$

Ő2) The suggestion that Methanothrix may carry out a methane oxidizing metabolism breaks with everything Őthat is known about this group, and the claim is not supported by any experimental data. This suggestion Őshould be removed.Ő Concerns 1&2: This manuscript is framed as a study that will draw significant insight from incubations. ŐIncubations with specific substrates or inhibitors can be very powerful tools in environmental Őmicrobiology, particularly when the microbial community responds to the incubation conditions, and when Őcare is taken to clearly describe the bulk geochemical processes that have occurred in the incubations. ŐUnfortunately, this is not the case in this study, while I understand that the bulk

BGD
of the description of the aĂŐincubations was previously published, a few key pieces of information have been left out of the current aAOmanuscript. It would likely appear to a reader that these are incubations in which methane oxidation is the Ődominant process since so much emphasis is put on AOM as compared to methanogenesis. AOM is the aAOmost discussed metabolism in the abstract, and a major conclusion is the surprising attribution of AOM Őmetabolism to Methanothrix. However, these incubations are NOT carrying out the net oxidation of aAOmethane, they are net methanogenic (see Figure 2b of Bar-Or 2017 "Positive methane concentrations Őreflect net methanogenesis during iron-coupled AOM."). To put the results more plainly: sequencing of aĂŐmethanogenic incubations reveals a dominant archaeon that is a well-known methanogen. When stated in Őthis way, I cannot support the publication of such a speculative assignment of AOM activity to ŐMethanothrix. The simplest explanation is that the dominant methanogen is growing via the dominant Őmethane cycling process, i.e. methanogenesis. The justification for any discussion of AOM relies heavily on aĂŐthe previous publication that found 13C methane was converted into 13C CO2, and this activity was Őinhibited by BES. Methanogens carry out backflux of isotopic label from methane to CO2, and the authors Őhave cited the classic paper that shows this (Zehnder and Brock, 1979). Methanothrix could indeed be Őresponsible for the conversion of 13C methane into 13C CO2, but this observation does not constitute aAÖevidence that they carry out net AOM in the environment or in these incubations. It is crucially important Őfor metabolisms that are so close to equilibrium for the authors to be very clear about whether they are Ősuggesting an organisms is making energy for growth by carrying out AOM, or whether the organism may Ősimply be responsible for the equilibration of isotope labels in the opposite direction of the process they address using for energy generation. Another line of evidence for AOM is reaction-diffusion modeling that was Őcarried out on Lake Kinneret sediments (Adler et al 2011), which concluded that there was peak Őmethanogenesis 5-12cm below the sediment surface, and there was deeper AOM region under that. But Ômicrobial 16s profiling carried out in Bar-Or et al 2015, did not show a sig-

**BGD**
nificant change of methanothrix ŐŐ(there referred to as methanosaeta) between the methanogenic and the methane oxidation zones. This is Őa big claim the authors are trying to make, and it would require some sort of direct evidence like: 1) if there Őwas an incubation where AOM was the dominant processes and the authors were able to show that Őmethanothrix was the only organism present with the seven step methanogenesis pathway; 2) or better Őyet that Methanothrix was enriched under these conditions vs. conditions without methane/Fe addition; 3) Őor, upon the addition of methane (and Fe?) there was a positive reaction of methanothrix based on Őmetatranscriptome analyses, 4) or, at the very least that in nature there was a correlation between Őmethanothrix abundance and the horizons where methane oxidation is occurring. Unfortunately, the Őcommunity did not significantly change under any incubation condition (line 45), and there is no correlation Őpresented from the natural distribution of species, so there is no valid justification for assigning a novel aĂŐrole to an organism that could just be making methane. Unless stronger evidence exists, all claims like the aĂŐone in line 375: "Our data hints that Methanothrix, which has not been considered to be involved in Fe-ŐAOM previously, has the potential to be involved in methane oxidation, as presented in figure 5" should be Őremoved. Ő

Response We thank the reviewer for this through discussion and fully agree and aware that in cases of Őincubations with net methanogenesis a plausible explanation for the involvement of Őmethanogens (not the bacteria of course) can be through a back flux of the methanogenesis Őprocess. Part of the work of our lab these days in several sets of incubations from different Ősettings is to try to separate between active ("true") AOM and back flux of methanogenesis, Őbut it is beyond the scope of this biological study. This point regarding the methanogens was Őprobably not clear and discussed enough in the original manuscript, and is clarified and Ődiscussed now in the revised version. Considering this, the methanogens that are involved in Őthe methane oxidation, in case it is back flux, can be indeed the main players in the system Őwhich increased with depths or incubation time. We agree that due
to the limited sample Ősize, statistical analyses of Bar-Or et al. 2015 results are impossible, but this study still Őshows a trend, suggesting an increase in the read abundance of Methanothrix with depth and Őtime. However, we agree that we need to be much more careful at this stage, and in the Őrevised text, we use very cautious language when considering the involvement of Őmethanogens in this process (we write now methanogenic archaea in general). Ő

Ő3) The authors do not carry out any calculations to support their claim that traditional ANME are not Őabundant enough to carry out the trace AOM they claim to observe, and no effort is made to engage with addet thermodynamic feasibility of the processes they are proposing, which is fairly straightforward and Őshould be done.Ő Concern 3: If the authors reject the isotope backflux idea (there is not a clear quantitative argument Őagainst this, even in Bar-Or et al 2017), and insist that there must be an organism subsisting on AOM in Őtheir incubations, then it is unclear why the minor, traditional ANME organisms will not suffice. In the Őabstract the authors write (lines 23-24) that "bonafide [sic] anaerobic oxidizers of methane (ANME) and Ődenitrifying methanotrophs Methylomirabilia (NC10) were scarce", discounting their role in AOM in these Ősediments. But then they highlight on line 25-26 "We show that putative aerobes, such as methane-Őoxidizing bacteria Methylomonas and their methylotrophic syntrophs methylotenera. . . can be involved in Ôthe oxidation of methane. . .". It is not at all clear why the authors feel that ANME should be discounted Őwhile aerobic methanotrophs should be accepted as being responsible for methane oxidation. On line 176 Őthe authors say that 0.3-0.8% of their reads map to ANME-1. And the very next paragraph the authors Ődiscuss the type I methanotrophs which are found to be 0.4-1.8% of the community. There is no Őmeaningful difference between 0.3-0.8% and 0.4- 1.8% in terms of abundance, so why do they feel Őcomfortable highlighting the possible role of aerobic methanotrophs at this abundance and not the Őanaerobic ones? Why have the aerobic methane oxidizers made it into Fig 5 but the bona fide ANME have Őnot? AOM is not the dominant process, so its seems reasonable that if there is a small methane oxidizing âÅÔcommu-

Interactive

comment

nity that it could be carried out by normal methane oxidizers that are in low abundance. The only aAOW to rule this out is to determine the rate of AOM, try to estimate what 0.3-0.8% read mapping may Őcorrespond to in terms of cell numbers, and then calculate a cell specific rate and show that this rate Őseems far too high when compared to values present in the literature for ANME rates. None of this work is Ôdone. When discussing possible metabolisms and their putative relative importance, it is very helpful to aĂŐdiscuss the thermodynamic feasibility of these reactions. But in the summary line 380-381 the authors Őwrite ". . .whether this process [methanothrix AOM] is justified from the thermodynamic and kinetic addperspectives, remains to be elucidated.". Doing the thermodynamic analysis should be a bare minimum Őreguirement when suggesting a remarkable new metabolism for an organism. What are the relative free aAOenergies associated with acetoclastic methanogenesis and then Fe-AOM vs. acetate oxidizing iron aAOreduction? For a study that is essentially just a single metagenomic analysis (since there is no noteworthy aAOdifference between any of the samples), the authors should at least attempt to supplement their discussion Őwith thermodynamic discussions. Ő

Response: We accept the comment, and based on our low AOM rates (~10-14 mol/cm3sec), ŐANME-1 may be indeed involved in the AOM process despite its low numbers, and we now Őstate it in the abstract and all along. Please note that the involvement of Methanothrix in ŐAOM has been also previously suggested, ŐŐ(https://www.sciencedirect.com/science/article/pii/S0048969720352062,

Őhttps://aem.asm.org/content/aem/83/11/e00645-17.full.pdf). Ő Regarding the thermodynamics, active Fe-AOM is a possible competitive process in this zone Őbased on calculations that were done already in our previous studies. In short, it can be Őshown that acetoclastic methanogenesis + Fe-AOM compared to acetoclastic iron reduction, Őor hydrogenotrophic methanogenesis (more dominant at this depth, (Adler, 2016)) + Fe-AOM Őcompared to hydrogenotrophic iron reduction result in more or less the same negative Gibbs Őenergy of around -200 kJ/mol (see the excel calculation in the attached file). We Őadded the thermodynamic
considerations to the revised version.Ő

To summarize our response to the major comments, we are not rejecting the role of the back  $\hat{a}\check{A}\check{O}$ flux. On the contrary, in our current lab work, we investigate it. Thus, we thank the reviewer  $\hat{a}\check{A}\check{O}$ for the strong suggestion and encouragement to discuss it also in this paper and to be more  $\hat{a}\check{A}\check{O}$ careful regarding the type of methanogens involved in methane oxidation. We, therefore,  $\hat{a}\check{A}\check{O}$ write "methanogenic archaea" instead of "Methanothrix" when discussing AOM. $\hat{a}\check{A}\check{O}$

Minor comments: Ő Ő "Consortium" should not be used interchangeably with "community" especially in the context of AOM Ő research where "consortium" is very commonly used to refers to a physical, presumably syntrophic Ő association between two microorganisms. Since no evidence is provided about actually association Ő between any organisms described in this study "consortium" should be replaced throughout with Ő Ő "community". Ő

Response: We replaced "consortium" with "community" as suggested.Ő

Line 361: "Our results show that in general, the phylogenetic diversity is a good predictor of the functional Ődiversity in these samples". This is too broad of a statement for a paper that has a fairly narrow focus on Őiron and methane cycling. Ő

Response: Although we highlight methane and iron cycling, we explored a wide array of Őfunctions (section "General metabolic potential", Fig. 2, Supplementary database 3). We, Őhowever, agree that this statement is not necessary and remove it. Ő

Line 20: not clear what "intrinsic" means in this context. Are any organisms in this sample not intrinsic?  $\hat{a}\check{A}\check{O}$

Response: We removed "intrinsic" as suggested.Ő

Line 63: Assigning Thermodesulfovibrio to a carbon oxidizing, iron reducing metabolism is wildly Őspeculative and should be removed unless more work is done to support the claim. The authors cite Spring Őet al 1993 (indirectly, by way of BarOr

BGD
et al 2015) for this claim. Spring et al does not make this claim, they Ösuggest as a throw-away hypothetical in the discussion section that it could be possible that ŐMagnetobacterium could gain energy from sulfide oxidation coupled to iron reduction. They had no Őevidence for that claim, just suggested it was possible because Magnetobacterium has magnetosomes and Őlives in sulfidic environments. If the authors want to follow up this speculation with analysis, then they Őcould look for the magnetosome genes in their metagenomes and see if they are phylogenetically aligned Őwith Magnetobacterium (see Lin et al 2014 for the genes in magnetobacterium, Őhttps://www.nature.com/articles/ismej201494). If these thermodesulfovibrio have magnetosomes then Őmaybe its worth mentioning this, but even then, it is probably worth noting that there is no actual evidence Őthat these organisms can grow in this way. Ő

Response: The ability of some Themodesulfovibiro to grow using iron as electron acceptor Őhas been shown experimentally – for example, Frank et al. 2016 indicate that: "Besides Ősulfate, strain N1 could also use sulfite, thiosulfate and Fe(III) as electron acceptors. However, Őgrowth with Fe(III) as electron acceptor was slow." ŐŐ(https://www.frontiersin.org/articles/10.3389/fmicb.2016.02000/full). T. yellowstonii was Őalso considered previously as a potential iron reducer ŐŐ(https://onlinelibrary.wiley.com/doi/abs/10.1111/gbi.12173). We added these citations to Őthe manuscript. Ő

Line 143-145: Here the use of "limiting nutrient" is confusing. This term often refers to something that is a Őgrowth requirement because it is needed for the production of biomolecules or cofactors, P, N, Fe, etc. ŐThis is a different concept than iron being used for the purpose of an electron acceptor, which seems to be Őthe focus of this study. Clarification is needed.Ő

Response: Thank you for pointing this out. To avoid this issue, we changed "nutrient" to ŐŐ"electron acceptor".Ő
Line 151: three groups are listed and then "3-6% read abundance, respectively". Incorrect usage of Őrespectively, not clear what each groups abundance is. Ő

Response: We removed "respectively" from this sentence.Ő

Line 158: class-level phylogenetic information should not be taken as evidence for the pH optimal for a Őgroup (the authors actually cite a paper that describes how a different species of thermodesulfovibrio is Őalkaliphilic as compared to other species in that genera). This is definitely is not evidence for acidic/basic Őmicroenvironments. Ő Response: Ő Please note that we suggest that Thermodesulfovibrio are either neutrophilic or al-kaliphilic. ŐWe now add an additional citation to Sekiguchi et al. 2008 ŐŐ(https://www.microbiologyresearch.org/content/journal/ijsem/10.1099/ijs.0.2008/000893ŐŐ-0#tab2), which shows pH optima between 6.5 and 7.5 for various Thermodesulfovibrio Őlineages. Candidatus Acidulodesulfobacterales, is often associated with pH <3 ŐŐ(https://www.nature.com/articles/s41396-019-0415-y). In this sentence, we used careful Őlanguage ("hints"), as we agree that our findings don't provide direct evidence for the Őpresence of microenvironments.Ő

Line 378: "positive correlation between Methanosarcinales abundance and concentrations of reduced iron  $\hat{a}$ ÅŐin the deep sediment sections (Bar-Or et al 2017)". This is a very strange claim and I cannot find any  $\hat{a}$ ÅŐsignificant data that supports it. Bar-Or 2017 does not include pore water profiles or depth profiles of  $\hat{a}$ ÅŐMethanosarcinales, so maybe this reference is supposed to be Bar-Or et al 2015? Even so, the data  $\hat{a}$ ÅŐpresented in Bar-Or et al 2015 Figure 4 is single replicate from three depth points. It looks like the  $\hat{a}$ ÅŐdifference between 6-9cm and 29-32cm for methanosarcinales is 50% -> 55% at most? With this level of  $\hat{a}$ ÅŐreplication this is not a significant correlation that should be taken as evidence supporting  $\hat{a}$ ÅŐmethanosarcinales being responsible for iron reduction. $\hat{a}$ ÅŐ

Response: ŐThank you for pointing out the mistake in the reference, this indeed
refers to Bar-Or et al., ŐŐ2015. As stated above, the number of samples in this study is indeed limited, yet a vertical Őgradient in the abundance of Methanothrix was observed. In general, this paragraph uses a Őnow very careful language, as mentioned above. Ő

Figure 4: something is wrong with the description, or the data presented. For OmcS LK-2017 the number Őnext to the bar is 4, which the caption says corresponds to the number of total reads mapped to a gene. ŐThat bar shows very fine delineations, "Deltaproteobacteria" is maybe 1/20th of the total area of the bar? ŐHow can you get 1/20th with only 4 reads mapped? This comment applies to other bars in the OmcS figure. ŐMaybe worth revisiting how these were calculated? Ő

Response: These numbers are normalized per million reads, we adjusted the legend accordingly.  $\hat{a} \tilde{A} \tilde{O}$

Line 389: "Another possible explanation for the methylated compound leakage is the reversibility of the Őenzymes involved in AOM, in particular methyl-CoM reductase". Mcr does not may methylated compounds Őlike the ones the authors are referring to in the forward or reverse direction, so the reversibility of this Őenzyme has nothing to do with this discussion. Ő Response: As suggested, we removed "in particular methyl-CoM reductase" from the sentence.Ő Figure 5. The schematic in the top left shows iron reduction (Fe(III) -> Fe(II)) producing electronsŐ Thank you for pointing this out, we adjusted Figure 5 so the electron is either transferred to ŐFe (III) or methanogens for methanogenesis. Ő

Please also note the supplement to this comment: https://bg.copernicus.org/preprints/bg-2020-329/bg-2020-329-AC5-supplement.zip

---

## Author Response (AR1)

Dear Prof. Tina Treude,

Thank you for giving us the opportunity to submit a revised version of our manuscript entitled "Metagenomic insights into the metabolism of microbial communities that mediate iron and methane cycling in Lake Kinneret iron-rich methanic sediments" to Biogeosciences. We appreciate the time and effort that you and the reviewers have dedicated to evaluate our paper and the constructive reviews and suggestions, which improved significantly the quality of our manuscript. We have carefully considered all notes and suggestions and revised the manuscript accordingly.
We hereby present point-by-point answers to the issues raised by all 4 reviewers (in bold). Please note that line numbers in our responses refer to the revised version of the manuscript.

We hope that the manuscript will now be suitable for publication in Biogeosciences.
Sincerely yours,
Michal Elul, on behalf of all co-authors.

Anonymous Referee #1

The paper addresses a topic of importance to readers of this journal: the microbial ecology of ferruginous sediments. The title is descriptive and therefore does not as clearly summarize the paper's major finding as a declarative title would, but it does accurately describe the paper's topic. The abstract provides a concise and complete summary. The paper is overall well-structured and clearly written, with fluent and precise language, and of appropriate length. The figures are of high quality. The findings largely confirm a previous study (Vuillemin et al 2018), and thus the findings overall are more confirmatory than novel, but important nonetheless. I have several suggestions for strengthening the methods and results as well as some missing citations: The paper includes metagenomic data on sediments incubated with various substrates for 470-days but never mentions specifics about the activities of these sediments for methane oxidation, iron reduction, methanogenesis, etc. Please summarize those geochemical data from the Bar-Or et al 2017 study at the start of the results section to set the stage for the metagenomics findings.

**We thank the referee for this observation and agree that a summary presenting the geochemical data on Bar-Or et al 2017 slurries is needed. In the revised version, as recommended, we devoted a section (3.1, lines 141-166) at the beginning of the results for this purpose. In this section, we describe the concentrations of relevant elements (methane, dissolved iron, manganese, nitrate, and sulfate) in the investigated sedimentary zone as well as the geochemical data on methane oxidation, iron reduction, and methanogenesis processes in Bar-Or et al 2017 slurries.**

2) My second main concern is regarding the methods and results for the PilA proteins, which were identified through a simple KEGG annotation without a detailed analysis necessary to confirm that the aromatic abundance and spacing was sufficient for predicted electroactivity. The authors should add that analysis, as in this paper (https://doi.org/10.1111/1758-2229.12809) to check that the PilA sequences contain the requisite cutoffs for predicted electroactivity (≥9.8% aromatic amino acids, ≤22  Raa aromatic gaps, and aromatic amino acids at ˇ residues 1, 24, 27, 50 and/or 51, and 32 and/or 57) because there are many other type IV-a pilin genes that can easily be mistaken as

electroactive PilA. A script is available for calculation of mature pilin length, percent aromatic amino acids and aromatic free gaps (https://github.com/GlassLabGT/Pythonscript) as described in this paper: https://doi.org/10.1111/1758-2229.12809. Also for the multiheme cytochromes, there are scripts available from a published study: 'cytochrome_stats.py' described in https://doi.org/10.3389/fmicb.2016.00913 and available at https://github.com/bondlab/scripts. Also, note that electroactive PilA are present in lineages outside of Deltaproteobacteria: see https://doi.org/10.1111/1758-2229.12809 https://doi.org/10.1038/ismej.2017.141 and https://doi.org/10.1128/mBio.00579-19

**As suggested, we confirmed that the aromatic abundance and spacing was sufficient for predicted electroactivity in the metagenome pilA sequences using the recommended script. We corrected Figure 4d, which now shows in the amended version of the manuscript, only the PilA open reading frames that correspond to the stringent parameters. Accordingly, we adjusted the text in this paragraph to: " The overall abundance of the MHC (secreted and trans-membranal), PilA and OmcS ORFs was 364-493, 35-45, 5-9 and 4-9 counts per million reads mapped, respectively. Our findings confirm that the phylogenetic diversity of microbes are capable of nanowire-mediated DIET extends beyond deltaproteobacterial lineages (Bray et al. 2020), as strict searches attributed pilA-like sequences not only to Desulfobacterota (Deltaproteobacteria), but also to Thermodesulfovibrionales, Burkholderiales, Gemmatimonadales, Aminicenantales, as well as WOR-3 and Firmicutes (Fig, 4d)".**
**We thank the reviewer for pointing out the 'cytochrome_stats.py' script, this will streamline our future analyses.**

3) As supplemental data, the authors should include FASTA files with the hits for each of the major genes discussed, so that readers can easily use the sequences, unless the metagenomes have been deposited in annotated form into NCBI. The NCBI BioProject does not contain any genomes with accessions to cite, so it is important for the FASTA files to be provided with the publication, or else there is no way for readers to locate the new sequences without reprocessing the raw metagenomes in the BioProject PRJNA637457 (indeed, there are no genomes listed on the BioProject page, so the data are hidden in SRAs, and not easily accessible for BLAST searches). Even better would be to include annotated metagenomes on NCBI and include the assigned NCBI accession numbers in the paper, but currently that is not simple except for metagenome assembled bins.

**As suggested by the reviewer, we submitted the metagenome to NCBI within the PRJNA637457 project. The metagenome is currently being processed and will be released ASAP. We supplemented the manuscript with amino acid sequences of the enzymes discussed (those involved in methanogenesis and extracellular electron transfer, as well as heterodisulfide reductase subunits) in FASTA files, referred to as Supplementary Database 6, 7, and 8. The above-mentioned FASTA files can be found at Figshare open accesses respiratory under the link - https://doi.org/10.6084/m9.figshare.c.5245157.v1. A separate link to each dataset is can be found in the amended version of the supplementary information.**

4) Consider citing papers by Kelly Wrighton's group on the importance of Candidatus Methanothrix paradoxum for methanogenesis in terrestrial sediments with oxygen exposure. For example: https://doi.org/10.1038/s41467-017-01753-4. Could also help

explain the occurrence of genes encoding oxygen-dependent methane mono oxygenases if there is occasional oxygen exposure in these sediments. Are they bioturbated?

**We thank the reviewer for this information. We assume that Lake Kineret sediments are not bioturbated in the depths that we examined (26-41cm). We now cite a paper from Kelly Wrighton's group (https://doi.org/10.1038/s41467-017-01753-4), lines 202-205: "Other notable archaeal lineages included the acetoclastic Methanothrix (1-3% read abundance), which are often found en masse in anoxic lake sediments (Smith and Ingram-Smith 2007; Schwarz et al. 2007; Carr et al. 2018) as well as in oxygenated soil, as recently discovered for Methanothrix paradoxum (Angle et al. 2017)".**

Specific comments:

Line 40-41: There has been quite a great deal of research on the diversity and metabolic potential of microbial communities in natural anoxic sediments over the past 40 years. I would not characterize this topic as "largely unknown". Please correct language here to focus on a more specific question, perhaps on ferruginous sediments.

**We agree with the reviewer that this line needed to be more focused on a specific topic. The text now reads, line 43-44 "However, the diversity and metabolic potential of the microbial communities in natural anoxic ferruginous sediments are not fully understood"**

L163-164: It is notable that Bathyarchaeia remained one of the dominant lineages even after sediment incubation. It is typical that Bathys quickly "die out" when sealed in bottles for a few weeks-months (for example, https://doi.org/10.1111/gbi.12239) and these were sealed for 470 days! The authors may want to attempt to culture Bathys out of these bottles, since they seem to be persisting, and perhaps even growing.

**Response: We are thrilled to try it!**

L205: change "anaerobic conditions" to "anoxic conditions" (metabolisms are anaerobic/aerobic; environments are oxic/anoxic) .

**Response: Corrected as suggested.**

L252: correct the misspelling of Methanosarcinales.

**Response: Corrected as suggested.**

L287: ORFs per what? Per metagenome?

**Response: Indeed, per metagenome, we added this clarification in the text line 334.**

Anonymous Referee #2

This study "Metagenomic insights into the metabolism of microbial communities that mediate iron and methane cycling in Lake Kinneret sediments" use metagenomics to investigated microbial communities associated with iron reduction and methane cycling from both natural Lake Kinneret sediments and iron amended slurry incubations. The data and interpretation is generally good. While I find the topic of this study certainly interesting for Biogeosciences, there are several aspects which should be addressed before publication. Lack of accompanying geochemical analysis, enzyme assay or transcripts analysis make the

study descriptive, mostly putative or based on prediction from reference database in results and discussion.

**We agree with the reviewer that a geochemical background and analyses of both the sedimentary zone and slurry incubations examined here are needed to be added (also noticed by reviewers 1,3 and 4). In the revised version, we added a full section (3.1) that address the geochemical aspect of the manuscript. Since our study was based on metagenomics, it can only raise hypotheses regarding the functionality of the studied communities. We agree that further experiments, such as enzyme assay or metatranscriptomics are needed to base our assumptions. We strongly believe, however, that this study provides a valuable basis for further investigation of Lake Kinneret communities and iron and methane metabolisms.**

Moreover, metagenomic analysis of four treatments shows not much different between them or at least the authors didn't present much difference, which question the experiment design or validity of method due to poor coverage of metagenomic method, especially when targeting a minor group in a complex sample.

**Albeit the overall similarities, we find some differences between the treatment, yet lack the statistical power to show them and often can only speculate regarding their nature. For example, BES additions appear to reduce the relative abundance of Methanosarcinales, but not Methanomicrobiales, as observed in the 16S rRNA amplicon read results. We agree that the small changes following the addition of BES are curious, yet at this point, we prefer not to overinterpret these changes. Iron mineral amendments may have little effect on the community structure, as iron is not limiting in these sediments. The overall similarity of the communities allowed us to increase the coverage and co-assemble the reads from the different libraries, being in our favor in this case. We believe that although the coverage was insufficient to cover the rare taxa in the way that high-quality bins could be assembled, metagenome-wide functional predictions and taxonomic assignments still provided important insights into this system.**

Metagenomics analysis only covers the ferruginous part of sediment core, so the title, abstract and descriptions throughout the text should be specific, rather than use "whole" lake sediment.

**We agree - in the revised version we emphasize in both the title, abstract and descriptions throughout the text that our analyses address only the deep iron-rich methanic part of the sediment in Lake Kinneret. The title has been changed to "Metagenomic insights into the metabolism of microbial communities that mediate iron and methane cycling in Lake Kinneret iron-rich methanic sediments"**

The names of microbes and genes should be in italic, first letter of proteins should be in Capital, please check and correct throughout the whole text

**Thank you for these observations, we made amendments throughout the text accordingly.**

Specific comments:
Line 35 "on average" and "up to" are redundant and not logical here, delete one.

**Corrected as suggested, line 37.**

Line 40 "largely unknown" is not precise here, actually there have many studies in recent years, in ferruginous sediments will be more specific.

**Response: This sentence now reads: "However, the diversity and metabolic potential of the microbial communities in natural anoxic ferruginous sediments are not fully understood."**

Line 46 change depleting to depleted

**Corrected as suggested, line 49.**

Line 71 Diversity of what?

**We refer to the diversity of bacteria and archaea. For clarity, this line now reads: "In all the treatments, the diversity of bacteria and archaea was similar to that of the natural sediments", line 76.**

Line 208-211 Did the author measured concentrations of H2 and SO4 in this study? Otherwise, they need to explain how they get these numbers.

**Our group measured these species. H2 concentrations were measured by Michal Adler and shown in her doctoral dissertation, and SO4 concentrations were measured in Adler et al. 2011; Sivan et al.2011 and Bar-Or et al., 2015. The references were added as suggested.**

Anonymous Referee #3

This manuscript on "Metagenomic insights into the metabolism of microbial communities that mediate iron and methane cycling in Lake Kinneret sediments" is very well written and organized. The title accurately describes the subject of the manuscript, though it is a bit dry and lacks any insight into what was concluded in the study. The abstract is clean and concise and effectively summarizes the key findings of the manuscript, which are largely descriptive.

**We thank the reviewer for the positive feedback.**

The introduction is also well constructed and (mostly) properly referenced, though the statement at line 40 of "largely unknown isn't exactly true.

**Following the recommendation of three out of the four reviewers, this line was changed to "However, the diversity and metabolic potential of the microbial communities in natural anoxic ferruginous sediments are not fully understood"**

However, my main concern with this paper is that there is no geochemical data from the incubations to confirm/support the metagenomic interpretations. The authors state at line 374 : "our geochemical experiments suggest: : :." however, no geochemical data is provided. As such, while the authors engage in thorough, well referenced discussion of inferred function based on homology searches, implying that there is experimental geochemical evidence to support their conclusions is misleading unless that data is presented. If it is available it needs to be presented, even if only in the supplement and not the focus of the main text.

**We thank the reviewer for this helpful comment. In the revised manuscript, we added a new section (3.1) that briefly addresses the geochemistry of the sampled sediments and**

the geochemical analyses of the slurry incubations. We supplement this discussion with figure S1 in the Supplement,  which shows the change in $\delta^{13}C$ of the DIC  of the slurry incubations, after Bar-Or et al.2017.

I find similarity between the in situ sediment samples and all of the incubations for which metagenomes are available to also be curious, especially in the presence of inhibitors. Perhaps some geochemical data could shed some light on this?

**We have observed some dissimilarities between the treatments, however, our analyses lack the statistical power to clearly define these differences. We can speculate that iron amendments had little effect on the composition of microbial communities, as iron is not a limiting factor in these sediments. Similarly, as we suspect that sulfate plays only a minor role in these sediments due to the low concentrations, the addition of molybdate may have only a negligible effect on the community structure.  Bar-Or et al. 2017 geochemical data (now presented as Supplementary Figure S1) show that the addition of BES completely halted methanotrophy and methanogenesis. We observed that the read abundance of some lineages, such as Methanosarcinales, declined in BES amendments (Supplementary Figure S2, S1 in the previous version). It is still unclear how other methanogens persist in BES-amended treatments, transcriptomics may elucidate this interesting phenomenon. It is important to note that the results here describe only the relative abundance. It is feasible that the cell numbers declined following the BES addition. In this study, the fact that the communities are similar among the treatments is, in fact, helpful for our analyses, allowing co-assembly and thus better genomic coverage.**

At line 71-72 the authors state that " slurry incubations: : :: : :produced substantial amounts of 13C-labelled DIC". How much is "substantial amounts"?

**We clarify this in the text and refer to the new Supplementary Figure 1: "These incubations, including a) $^{13}CH_4$, b) $^{13}CH_4$ + Hematite, or c) $^{13}CH_4$ + amorphous iron + molybdate (A.Fe(III)+MoO$_4$) produced substantial amounts of $^{13}$C-labelled dissolved inorganic carbon over 470 days (80-450‰, Fig. S1 in the Supplement).". As stated above, we added section 3.1 to introduce the geochemical data.**

Was there iron reduction? H2 production? Or did the slurry just sit there static and are just a reflection of the initial sediment slurry sitting there for over a year, as it sort of looks like from the non-departure from the t0 microbial community (Figure S2).

**Iron reduction occurred in the slurry incubations. We address this subject in the newly added section 3.1 , Lines 159-160 " Ferrous iron concentrations increased by ~20–50 µM following iron oxide amendments (with and without molybdate addition), indicating that iron was reduced." Unfortunately, H2 was not measured in the slurry incubations.**

There seems to be some presentation of in situ geochemical data (lines 208-209) though it's unclear if this was measured or a previously reported value.

**The values mentioned here are previously reported values. To clarify this issue, the respective references we added: The hydrogen concentration in the Fe-AOM horizon is ~20 µM gr-1 sediment  (Adler 2015). Given that sulfate is below the detection limit there (<10µM, Adler et al., 2011, Sivan et al., 2011), hydrogen scavenging may also be coupled to metal reduction, most likely by Deltaproteobacterial lineages, some of which may be syntrophic (e.g. Syntrophobacterales). "**

The manuscript by Elul et al reports the results of 16s amplicon and shotgun metagenomic analysis of a narrow sediment horizon from Lake Kinneret. These DNA analyses were conducted on freshly sampled sediment and sediment that had undergone the incubations characterized in detail in Bar-Or et al 2017. The authors focus their attention on enzyme systems that may be associated with iron or methane cycling. The authors provide information on the phylogenetic composition of the microbial community in general, as well as assign phylogenetic composition to specific enzymes by BLASTing the metagenome reads against the RefSeq database.

**We thank the reviewer for this thorough review.**

Major concerns:
1) Insufficient information is given about the incubations which is needed to fully evaluate the likelihood of the conclusions presented in the current work (most crucially, these incubations are methanogenic).

**As requested by all the referees, we added section 3.1, named "Geochemical evidence for iron coupled AOM in Lake Kinneret iron-rich methanic sediments". In this section, we describe the change in ferrous iron, δ13CDIC and methane concentrations with time in the incubations. This section also includes a description of the concentrations of relevant elements (methane, dissolved iron, manganese, nitrate, and sulfate) in this investigated sedimentary methanogenic zone. We added also that these incubations are indeed methanogenic (see more below).**

2) The suggestion that Methanothrix may carry out a methane oxidizing metabolism breaks with everything that is known about this group, and the claim is not supported by any experimental data. This suggestion should be removed.

Concerns 1&2: This manuscript is framed as a study that will draw significant insight from incubations. Incubations with specific substrates or inhibitors can be very powerful tools in environmental microbiology, particularly when the microbial community responds to the incubation conditions, and when care is taken to clearly describe the bulk geochemical processes that have occurred in the incubations. Unfortunately, this is not the case in this study, while I understand that the bulk of the description of the incubations was previously published, a few key pieces of information have been left out of the current manuscript. It would likely appear to a reader that these are incubations in which methane oxidation is the dominant process since so much emphasis is put on AOM as compared to methanogenesis. AOM is the most discussed metabolism in the abstract, and a major conclusion is the surprising attribution of AOM metabolism to Methanothrix. However, these incubations are NOT carrying out the net oxidation of methane, they are net methanogenic (see Figure 2b of Bar-Or 2017 "Positive methane concentrations reflect net methanogenesis during iron coupled AOM."). To put the results more plainly: sequencing of methanogenic incubations reveals a dominant archaeon that is a well-known methanogen. When stated in this way, I cannot support the publication of such a speculative assignment of AOM activity to Methanothrix. The simplest explanation is that the dominant methanogen is growing via the dominant methane cycling process, i.e. methanogenesis. The justification for any discussion of AOM relies heavily on the previous publication that found 13C methane was converted into 13C CO2, and this activity was inhibited by BES. Methanogens carry out backflux of isotopic label from methane to CO2, and the authors have cited the classic paper that shows this (Zehnder and Brock, 1979). Methanothrix could indeed be responsible for the

conversion of 13C methane into 13C CO2, but this observation does not constitute evidence that they carry out net AOM in the environment or in these incubations. It is crucially important for metabolisms that are so close to equilibrium for the authors to be very clear about whether they are suggesting an organisms is making energy for growth by carrying out AOM, or whether the organism may simply be responsible for the equilibration of isotope labels in the opposite direction of the process they are using for energy generation. Another line of evidence for AOM is reaction-diffusion modeling that was carried out on Lake Kinneret sediments (Adler et al 2011), which concluded that there was peak methanogenesis 5-12cm below the sediment surface, and there was deeper AOM region under that. But microbial 16s profiling carried out in Bar-Or et al 2015, did not show a significant change of methanothrix (there referred to as methanosaeta) between the methanogenic and the methane oxidation zones. This is a big claim the authors are trying to make, and it would require some sort of direct evidence like: 1) if there was an incubation where AOM was the dominant processes and the authors were able to show that methanothrix was the only organism present with the seven step methanogenesis pathway; 2) or better yet that Methanothrix was enriched under these conditions vs. conditions without methane/Fe addition; 3) or, upon the addition of methane (and Fe?) there was a positive reaction of methanothrix based on metatranscriptome analyses, 4) or, at the very least that in nature there was a correlation between methanothrix abundance and the horizons where methane oxidation is occurring. Unfortunately, the community did not significantly change under any incubation condition (line 45), and there is no correlation presented from the natural distribution of species, so there is no valid justification for assigning a novel role to an organism that could just be making methane. Unless stronger evidence exists, all claims like the one in line 375: "Our data hints that Methanothrix, which has not been considered to be involved in Fe-AOM previously, has the potential to be involved in methane oxidation, as presented in figure 5" should be removed.

**We thank the reviewer for this through discussion and fully agree and aware that in cases of incubations with net methanogenesis a plausible explanation for the involvement of methanogens (not the bacteria of course) can be through a back flux of the methanogenesis process. Part of the work of our lab these days in several sets of incubations from different settings is to try to separate between active ("true") AOM and back flux of methanogenesis, but it is beyond the scope of this biological study. This point regarding the methanogens was probably not clear and discussed enough in the original manuscript, and is clarified and discussed now in the revised version. Considering this, the methanogens that are involved in the methane oxidation, in case it is back flux, can be indeed the main players in the system which increased with depths or incubation time. We agree that due to the limited sample size, statistical analyses of Bar-Or et al. 2015 results are impossible, but this study still shows a trend, suggesting an increase in the read abundance of Methanothrix with depth and time.**
**However, we agree that we need to be much more careful at this stage, and in the revised text, we use very cautious language when considering the involvement of methanogens in this process (we write now methanogenic archaea in general).**

3) The authors do not carry out any calculations to support their claim that traditional ANME are not abundant enough to carry out the trace AOM they claim to observe, and no effort is made to engage with the thermodynamic feasibility of the processes they are proposing, which is fairly straightforward and should be done.
Concern 3: If the authors reject the isotope backflux idea (there is not a clear quantitative argument against this, even in Bar-Or et al 2017), and insist that there must be an organism subsisting on AOM in their incubations, then it is unclear why the minor, traditional ANME

organisms will not suffice. In the abstract the authors write (lines 23-24) that "bonafide [sic] anaerobic oxidizers of methane (ANME) and denitrifying methanotrophs Methylomirabilia (NC10) were scarce", discounting their role in AOM in these sediments. But then they highlight on line 25-26 "We show that putative aerobes, such as methane-oxidizing bacteria Methylomonas and their methylotrophic syntrophs methylotenera. . . can be involved in the oxidation of methane. . .". It is not at all clear why the authors feel that ANME should be discounted while aerobic methanotrophs should be accepted as being responsible for methane oxidation. On line 176 the authors say that 0.3-0.8% of their reads map to ANME-1. And the very next paragraph the authors discuss the type I methanotrophs which are found to be 0.4-1.8% of the community. There is no meaningful difference between 0.3-0.8% and 0.4- 1.8% in terms of abundance, so why do they feel comfortable highlighting the possible role of aerobic methanotrophs at this abundance and not the anaerobic ones? Why have the aerobic methane oxidizers made it into Fig 5 but the bona fide ANME have not? AOM is not the dominant process, so its seems reasonable that if there is a small methane oxidizing community that it could be carried out by normal methane oxidizers that are in low abundance. The only way to rule this out is to determine the rate of AOM, try to estimate what 0.3-0.8% read mapping may correspond to in terms of cell numbers, and then calculate a cell specific rate and show that this rate seems far too high when compared to values present in the literature for ANME rates. None of this work is done. When discussing possible metabolisms and their putative relative importance, it is very helpful to discuss the thermodynamic feasibility of these reactions. But in the summary line 380-381 the authors write ". . .whether this process [methanothrix AOM] is justified from the thermodynamic and kinetic perspectives, remains to be elucidated.". Doing the thermodynamic analysis should be a bare minimum requirement when suggesting a remarkable new metabolism for an organism. What are the relative free energies associated with acetoclastic methanogenesis and then Fe-AOM vs. acetate oxidizing iron reduction? For a study that is essentially just a single metagenomic analysis (since there is no noteworthy difference between any of the samples), the authors should at least attempt to supplement their discussion with thermodynamic discussions.

**We accept the comment, and based on our low AOM rates (~10-14 mol/cm3sec), ANME-1 may be indeed involved in the AOM process despite its low numbers, and we now state it in the abstract and all along. Please note that the involvement of Methanothrix in AOM has been also previously suggested, (https://www.sciencedirect.com/science/article/pii/S0048969720352062, https://aem.asm.org/content/aem/83/11/e00645-17.full.pdf).**
**Regarding the thermodynamics, active Fe-AOM is a possible competitive process in this zone based on calculations that were done already in our previous studies. In short, it can be shown that acetoclastic methanogenesis + Fe-AOM compared to acetoclastic iron reduction, or hydrogenotrophic methanogenesis (more dominant at this depth, (Adler, 2016)) + FeAOM compared to hydrogenotrophic iron reduction result in more or less the same negative Gibbs energy of around -200 kJ/mol. We added the thermodynamic considerations to the revised version in the text and as Table S2 in the supplementary material.**
**To summarize our response to the major comments, we are not rejecting the role of the back flux. On the contrary, in our current lab work, we investigate it. Thus, we thank the reviewer for the strong suggestion and encouragement to discuss it also in this paper and to be more careful regarding the type of methanogens involved in methane oxidation. We, therefore, write "methanogenic archaea" instead of "Methanothrix" when discussing AOM.**

Minor comments:
"Consortium" should not be used interchangeably with "community" especially in the context of AOM research where "consortium" is very commonly used to refers to a physical, presumably syntrophic association between two microorganisms. Since no evidence is provided about actually association between any organisms described in this study "consortium" should be replaced throughout with "community".

**We replaced "consortium" with "community" as suggested.**

Line 361: "Our results show that in general, the phylogenetic diversity is a good predictor of the functional diversity in these samples". This is too broad of a statement for a paper that has a fairly narrow focus on iron and methane cycling.

**Although we highlight methane and iron cycling, we explored a wide array of functions (section "General metabolic potential", Fig. 2, Supplementary database 3). We, however, agree that this statement is not necessary and remove it.**

Line 20: not clear what "intrinsic" means in this context. Are any organisms in this sample not intrinsic?

**We removed "intrinsic" as suggested.**

Line 63: Assigning Thermodesulfovibrio to a carbon oxidizing, iron reducing metabolism is wildly speculative and should be removed unless more work is done to support the claim. The authors cite Spring et al 1993 (indirectly, by way of BarOr et al 2015) for this claim. Spring et al does not make this claim, they suggest as a throw-away hypothetical in the discussion section that it could be possible that Magnetobacterium could gain energy from sulfide oxidation coupled to iron reduction. They had no evidence for that claim, just suggested it was possible because Magnetobacterium has magnetosomes and lives in sulfidic environments. If the authors want to follow up this speculation with analysis, then they could look for the magnetosome genes in their metagenomes and see if they are phylogenetically aligned with Magnetobacterium (see Lin et al 2014 for the genes in magnetobacterium, https://www.nature.com/articles/ismej201494). If these thermodesulfovibrio have magnetosomes then maybe its worth mentioning this, but even then, it is probably worth noting that there is no actual evidence that these organisms can grow in this way.

**The ability of some Themodesulfovibiro to grow using iron as electron acceptor has been shown experimentally – for example, Frank et al. 2016 indicate that: "Besides sulfate, strain N1 could also use sulfite, thiosulfate and Fe(III) as electron acceptors. However, growth with Fe(III) as electron acceptor was slow." (https://www.frontiersin.org/articles/10.3389/fmicb.2016.02000/full). T. yellowstonii was also considered previously as a potential iron reducer (https://onlinelibrary.wiley.com/doi/abs/10.1111/gbi.12173). We added these citations to the manuscript.**

Line 143-145: Here the use of "limiting nutrient" is confusing. This term often refers to something that is a growth requirement because it is needed for the production of biomolecules or cofactors, P, N, Fe, etc. This is a different concept than iron being used for

the purpose of an electron acceptor, which seems to be the focus of this study. Clarification is needed.

**Thank you for pointing this out. To avoid this issue, we changed "nutrient" to "electron acceptor".**

Line 151: three groups are listed and then "3-6% read abundance, respectively". Incorrect usage of respectively, not clear what each groups abundance is.

**We removed "respectively" from this sentence.**

Line 158: class-level phylogenetic information should not be taken as evidence for the pH optimal for a group (the authors actually cite a paper that describes how a different species of thermodesulfovibrio is alkaliphilic as compared to other species in that genera). This is definitely is not evidence for acidic/basic microenvironments.

**Please note that we suggest that Thermodesulfovibrio are either neutrophilic or alkaliphilic. We now add an additional citation to Sekiguchi et al. 2008 (https://www.microbiologyresearch.org/content/journal/ijsem/10.1099/ijs.0.2008/00089 3-0#tab2), which shows pH optima between 6.5 and 7.5 for various Thermodesulfovibrio lineages. Candidatus Acidulodesulfobacterales, is often associated with pH <3 (https://www.nature.com/articles/s41396-019-0415-y). In this sentence, we used careful language ("hints"), as we agree that our findings don't provide direct evidence for the presence of microenvironments.**

Line 378: "positive correlation between Methanosarcinales abundance and concentrations of reduced iron in the deep sediment sections (Bar-Or et al 2017)". This is a very strange claim and I cannot find any significant data that supports it. Bar-Or 2017 does not include pore water profiles or depth profiles of Methanosarcinales, so maybe this reference is supposed to be Bar-Or et al 2015? Even so, the data presented in Bar-Or et al 2015 Figure 4 is single replicate from three depth points. It looks like the difference between 6-9cm and 29-32cm for methanosarcinales is 50% -> 55% at most? With this level of replication this is not a significant correlation that should be taken as evidence supporting methanosarcinales being responsible for iron reduction.

**Thank you for pointing out the mistake in the reference, this indeed refers to Bar-Or et al., 2015. As stated above, the number of samples in this study is indeed limited, yet a vertical gradient in the abundance of Methanothrix was observed. In general, this paragraph uses a now very careful language, as mentioned above.**

Figure 4: something is wrong with the description, or the data presented. For OmcS LK-2017 the number next to the bar is 4, which the caption says corresponds to the number of total reads mapped to a gene. That bar shows very fine delineations, "Deltaproteobacteria" is maybe 1/20th of the total area of the bar? How can you get 1/20th with only 4 reads mapped? This comment applies to other bars in the OmcS figure. Maybe worth revisiting how these were calculated?

**These numbers are normalized per million reads, we adjusted the legend accordingly.**

Line 389: "Another possible explanation for the methylated compound leakage is the

reversibility of the enzymes involved in AOM, in particular methyl-CoM reductase". Mcr does not may methylated compounds like the ones the authors are referring to in the forward or reverse direction, so the reversibility of this enzyme has nothing to do with this discussion.

**As suggested, we removed "in particular methyl-CoM reductase" from the sentence.**

Figure 5. The schematic in the top left shows iron reduction (Fe(III) -> Fe(II)) producing electrons

**Thank you for pointing this out, we adjusted Figure 5 so the electron is either transferred to Fe (III) or methanogens for methanogenesis.**